# Epigenome-wide association study of serum urate reveals insights into urate co-regulation and the *SLC2A9* locus

Elevated serum urate levels, a complex trait and major risk factor for incident gout, are correlated with cardiometabolic traits via incompletely understood mechanisms. DNA methylation in whole blood captures genetic and environmental influences and is assessed in transethnic meta-analysis of epigenome-wide association studies (EWAS) of serum urate (discovery, $n = 12,474$, replication, $n = 5522$). The 100 replicated, epigenome-wide significant ($p < 1.1E–7$) CpGs explain 11.6% of the serum urate variance. At *SLC2A9*, the serum urate locus with the largest effect in genome-wide association studies (GWAS), five CpGs are associated with *SLC2A9* gene expression. Four CpGs at *SLC2A9* have significant causal effects on serum urate levels and/or gout, and two of these partly mediate the effects of urate-associated GWAS variants. In other genes, including *SLC7A11 and PHGDH*, 17 urate-associated CpGs are associated with conditions defining metabolic syndrome, suggesting that these CpGs may represent a blood DNA methylation signature of cardiometabolic risk factors. This study demonstrates that EWAS can provide new insights into GWAS loci and the correlation of serum urate with other complex traits.

Serum urate concentrations represent a complex genetic trait, influenced by genetic variation in many genomic regions[1–3], and by environmental influences such as intake of purine-rich food, alcohol, fructose, and certain diuretic medications[4,5]. Elevated serum urate levels, hyperuricemia, can lead to mono-sodium urate crystal deposition and thereby cause gout[6], the most common type of inflammatory arthritis among adults. Moreover, observational studies show that serum urate levels are associated with various cardiometabolic risk factors including hypertension and obesity[7–9]. Gaining insights into the regulation of serum urate levels is important not only to uncover targets to develop urate-lowering and anti-inflammatory therapies to treat gout, but may also provide insights into shared pathways with cardiometabolic risk factors.

Previous studies reported the estimated heritability of serum urate levels in the general population as 30–70%[3,10,11]. Large-scale genome-wide association studies (GWAS) of serum urate levels have identified urate-associated genetic variants at >200 loci[1–3,12]. Genes in the identified loci suggest two major themes: the first is related to urate transport in the kidney and intestine, which determines net urate excretion, and the second is related to genetic co-regulation. The genes in loci most strongly associated with serum urate levels encode for urate transporters (e.g., SLC2A9, SLC22A12, ABCG2, SLC22A11, SLC17A1) or of sub-strates that may be exchanged for urate (e.g., SLC16A9), as well as for regulatory proteins of urate transporters (e.g., PDZK1)[13,14]. While the underlying causal variant rs2231142 at ABCG2, a missense variant causing reduced function, has been identified[15], the causal mechanisms underlying the strongest genetic associa-tion signal with serum urate, which maps to the urate transporter gene SLC2A9, remain largely unknown[16].

Regarding the potential genetic co-regulation between serum urate levels and multiple cardiometabolic risk factors such as triglyceride levels, blood pressure, obesity, and insulin resistance, we observed high genetic correlation between these risk factors and serum urate based on genome-wide genetic association statistics[3]. Such genetic co-regulation could explain the observed association of serum urate and cardiometabolic risk factors from epidemiological studies. It may, at least in part, be mediated by transcription factors with major regulatory roles in both liver and kidney such as HNF1A and HNF4A[3]. Genetic variants in these transcription factors are associated not only with serum urate levels, but also with impaired glucose handling and type 2 dia-betes mellitus as well as serum cholesterol and triglyceride levels[17–19], suggesting that coordinated gene regulation may have joint effects on serum urate and hepatic metabolism.

DNA methylation is an important mechanism of gene regulation and may reveal new insights into the biological processes that influence serum urate levels. We thus performed epigenome-wide association studies (EWAS) of serum urate levels with two scientific aims: first, to detect CpGs associated with serum urate levels and to investigate whether differential methylation may connect genetic risk variants of unknown molecular mechanism with serum urate levels. Second, to evaluate whether urate-associated differentially methylated CpGs are associated with cardiometabolic risk factors, pointing toward shared regulation of the implicated genes.

Here we perform meta-analyses of EWAS of serum urate levels followed by replication, using data from 17,996 individuals from four ancestry groups. Downstream characterization of urate-associated CpGs includes association with differential gene expression, Mendelian randomization (MR), and mediation as well as enrichment analyses. We show that significantly associated CpGs explain 11.6% of the serum urate variance in a dataset not included in the EWAS. Differential methylation of CpGs in SLC2A9 has causal effects on serum urate levels and mediates the effect of known urate-associated genetic variants. Urate-associated CpGs show associations with several cardiometabolic traits, con-sistent with their observational relationships with serum urate. Our study generates both independent and complementary insights to those obtained from GWAS of serum urate levels.

## Results

**Characteristics of the study participants**. The discovery analysis included up to 12,474 participants from 16 cohorts (European ancestry [EA]: 6968, African Americans [AA]: 2101, South Asian ancestry [SA]: 2720, and Sub-Saharan Africans [SSA]: 685). The replication analysis included 5522 participants from 8 cohorts with one study (the Normative Aging Study) including only men (EA: 3338, AA: 2184). Across these cohorts, the median of the average age within each cohort was 56 (25th, 75th percentile: 51, 63), the median of the proportion of men was 47% (25th, 75th percentile: 39%, 49%), and the median of mean serum urate levels was 5.3 mg/dL (25th, 75th percentile: 5.1, 5.6; Supplementary Data 1 and Supplementary Note 1). Study-specific methods and information on blood cell type proportions used to estimate associations between urate and DNA methylation independent of cell type composition are reported in the Methods section and Supplementary Data 2 and 3. A flowchart of the meta-analyses and follow-up characterization is presented in Fig. 1.

**Discovery and replication of urate-associated CpGs**. The dis-covery analysis identified 140 significant CpGs ($p < 1.1E–7 = 0.05/441,854$ CpGs analyzed), of which 100 replicated (consistent effect direction, $p < 0.05$ in the replication analysis, and overall meta-analysis $p < 1.1E–7$; Supplementary Data 4 and Fig. 2). The estimated inflation factor of the meta-analysis combining discovery and repli-cation was 1.06 (Supplementary Fig. 1). The study-specific results had a median inflation factor of 0.99 (min: 0.86, max: 1.06, Supplemen-tary Data 2), and were corrected prior to any meta-analysis when >1. The heterogeneity of most replicated CpGs was low to moderate: the median $I^2$ from the meta-analysis of the 17 EA studies was 11% (25th, 75th percentile: 0.0, 31.3), from the five AA studies was 0% (25th, 75th percentile: 0.0, 35.1), and from the overall meta-analysis of the four ancestry groups was 30.2% (25th, 75th percentile: 0.0, 60.0, Supplementary Data 4). Of the 32 CpGs that showed hetero-geneity >50% in the meta-analysis of the four ancestry groups, 90% (29 CpGs) had effect estimates in the same direction across the ancestry groups (Supplementary Fig. 2A–AF). Ancestry-specific results of CpGs with $p < 1.1E–7$ are reported in Supplementary Data 5 [EA], 6 [AA], and 7 [SA] if present, and with $p < 1E–5$ otherwise (Supplementary Data 8 [SSA]).

The follow-up analyses focused on the replicated CpGs from the overall meta-analysis, in which the CpG with the lowest $p$ value was at SLC7A11 (cg06690548, $p = 2.18E–59$). Among the ten CpGs with the lowest $p$ values, two mapped to PHGDH (cg14476101, $p = 5.87E–38$; cg16246545, $p = 1.77E–24$), which encodes for 3-phosphoglycerate dehydrogenase, and five to SLC2A9 (cg21795255, cg20479063, cg11266682, cg00071950, cg13841979; $p < 1E–25$), which encodes the urate transporter GLUT9 and is the locus with the largest effect size in GWAS of serum urate[3]. The probe cg21795255 at SLC2A9 had a common SNP at the extension base of the DNA methylation probe (rs3796839, minor allele frequency of 46% in GnomAD)[20] and was excluded from all follow-up analyses, which therefore included 99 replicated CpGs annotated to 81 genes. Of these, 24 genes contained replicated CpGs with significant findings from our follow-up analyses and are featured in Table 1.

**Heritability of urate-associated CpGs and explained serum urate variance**. DNA methylation can be influenced by genetics and environmental factors[21]. Heritability of DNA methylation levels at a

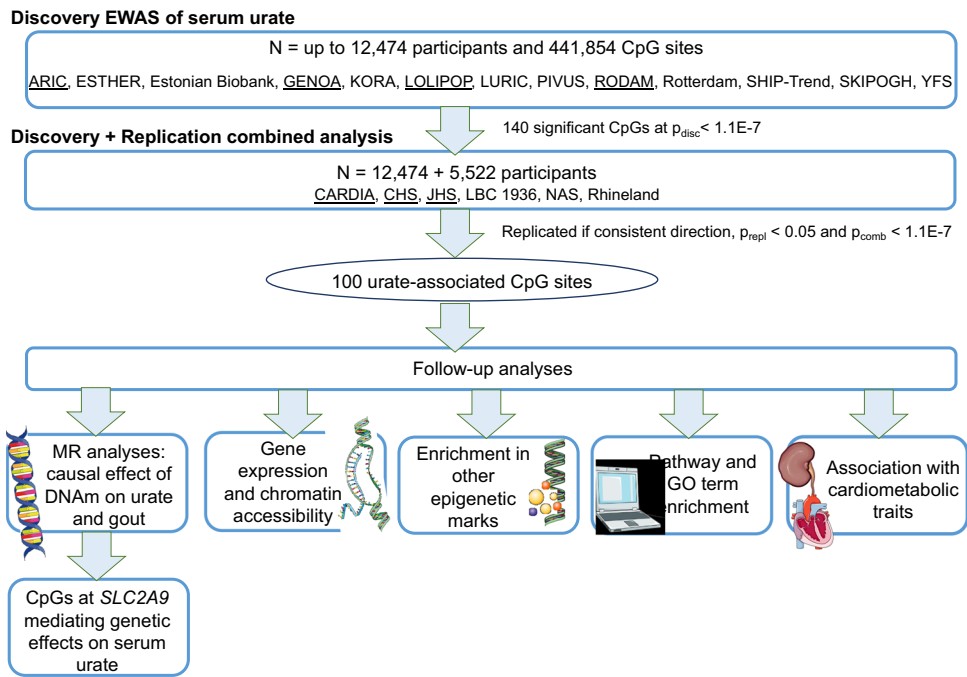

**Fig. 1 Flowchart of analyses.** MR Mendelian randomization, GO Gene Ontology. Icons were downloaded from smart.servier.com under the Creative Commons Attribution 3.0 Unported License.

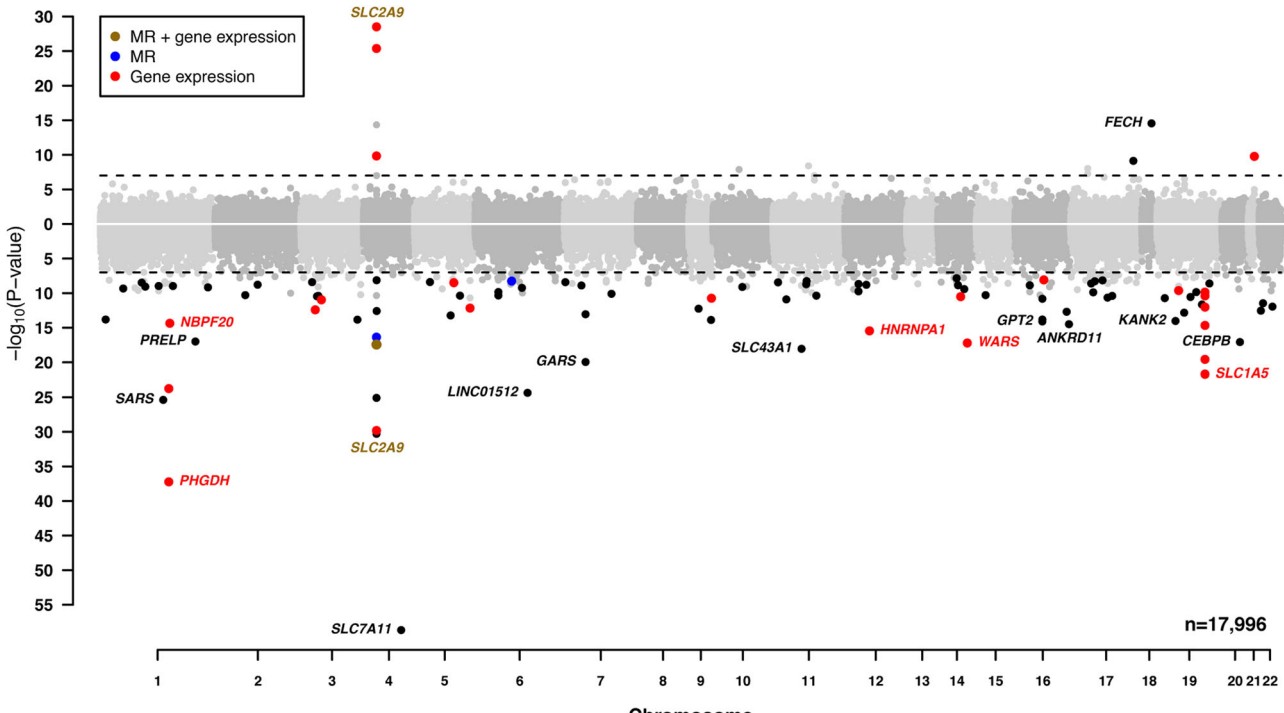

**Fig. 2 CpGs in the EWAS of serum urate from the combined meta-analysis of discovery and replication results (*n* = 17,996).** The CpGs are ordered by their chromosomal position on the *x*-axis with their −log₁₀(*p* value) of the association on the *y*-axis. CpGs with positive and negative effect estimates are plotted in the upper and lower panels, respectively. The dotted horizontal lines represent the level of significance corrected for multiple testing (two-sided *p* < 1.1E−7). Black: replicated CpGs; brown: gene with replicated CpGs associated with gene expression in monocytes and significant causal effects for serum urate; blue: replicated CpGs with significant causal effects on serum urate or gout; red: replicated CpGs associated with gene expression in monocytes or whole blood. Gene names are displayed if the gene had at least one CpG with *p* value < 1E−14. MR Mendelian randomization.

**Table 1 Genes having replicated CpG(s) associated with gene expression in whole blood or monocytes, with significant causal effects for urate or gout, associated with cardiometabolic traits, or annotated as a GWAS locus for urate or gout.**

| Chr | Gene | Function of encoded protein | No. of replicated CpG(s) | Direction of effect of the replicated CpGs | | | Cardiometabolic traits | Causal effect | GWAS locus (urate, gout) |
|---|---|---|---|---|---|---|---|---|---|
| | | | | Serum urate | Gene expression in whole blood | Gene expression in monocyte | | | |
| 1 | LRRC8D | Anion channel component | 1 | → | | → | | | |
| 1 | NBPF20/ PDZK1 | Neuroblastoma breakpoint family member/ scaffolding protein incl. urate transporters | 1 | → | | | → | | Urate, gout |
| 1 | PHGDH | Enzyme in L-serine biosynthesis | 2 | → | → | → | → | | |
| 1 | SARS | Enzyme catalyzing L-serine transfer to tRNA | 1 | → | | → | → | | |
| 3 | CLEC3B | Encodes tetranectin | 1 | → | → | → | | | |
| 3 | PARP3 | Encodes poly(ADP-ribosyl)transferase 3 | 1 | ↑→ | → | ↑→ | → | | |
| 4 | SLC2A9a | Urate transporter | 7 | ↑→ | | ↑→ | → | Urate, gout | Urate, gout |
| 4 | SLC7A11 | Component of cysteine/glutamate exchanger | 1 | → | | ← | | | |
| 5 | PURA | DNA-binding protein | 1 | → | | | → | | |
| 6 | LINC01512 | RNA gene | | → | | | → | | |
| 7 | GARS | Enzyme catalyzing glycine transfer to tRNA | 2 | → | | | → | | |
| 7 | SH2B2 | Adapter protein for tyrosine kinase receptor family members | 1 | → | | | → | | |
| 9 | UAP1L1 | Encodes UDP-N-Acetylglucosamine Pyrophosphorylase 1 Like 1 | 1 | → | ← | ← | | | |
| 11 | CPT1A | Enzyme in mitochondrial uptake of long-chain fatty acids | 1 | → | | | → | | |
| 11 | HRASLS2 | (PLAAT2) enzyme with phospholipase A1/2 and acyltransferase activities | 1 | → | | | | Urate | |
| 11 | SLC43A1 | Large neutral amino acid transporter | 1 | → | | | → | | |
| 11 | WEE1 | Ser/Thr tyrosine kinase | 1 | → | | | → | | |
| 14 | GALC | Enzyme in hydrolyzation of glycolipids | 1 | → | | → | | | |
| 14 | WARS | Enzyme catalyzing tryptophan transfer to tRNA | 1 | → | → | → | | | |
| 16 | NOD2 | Protein involved in immune response | 1 | → | | ← | → | | |
| 19 | BCL3 | Transcriptional activator | 1 | → | | ← | → | | |
| 19 | NFIX | Transcription factor | 1 | → | | ← | | | |
| 19 | SLC1A5 | Neutral amino acid transporter | 6 | → | → | → | → | | |
| 21 | ABCG1 | Member of the ABC transporter family, phospholipid efflux | 1 | ← | → | → | → | | |

The arrows indicate the effect direction of DNA methylation at each replicated CpG.
Gene function was obtained from Uniprot, EntrezGene, and/or PubMed.
Chr chromosome.
aAt SLC2A9, the up and down arrows indicate that some CpGs in this gene were associated with serum urate and gene expression in the positive direction and some in the negative direction.

CpG estimates the proportion of variance in DNA methylation levels at the CpG that can be attributed to additive genetics; it does not refer to germline inheritance of DNA methylation. Across three datasets from populations of EA, the heritability estimates of the replicated CpGs varied (Supplementary Data 9). The mean heritability estimates for the 99 replicated, urate-associated CpGs was higher than the mean heritability across all CpGs assessed in each of the three studies: 0.36 (urate-associated CpGs) vs. 0.16 (HM450K array) based on Hannon et al.[22], 0.54 vs. 0.19 for van Dongen et al.[23], and 0.52 vs. 0.19 for McRae et al.[24]. Among the seven replicated sites at *SLC2A9*, the mean heritability combining the three datasets ranged from 0.39 at cg03725404 to 0.93 at cg11266682. These observations suggest that the DNA methylation at some of the urate-associated CpGs may reflect the effect of common genetic variants on serum urate levels.

The proportion of serum urate variance explained by the replicated, urate-associated CpGs was estimated in a separate study sample using the coefficient of determination ($R^2$) obtained from linear regression (Methods). Compared to an age- and sex-adjusted model, the replicated CpGs explained an additional 11.6% of the serum urate variance. In a recent GWAS of serum urate based on data from 457,690 individuals, replicated genetic index variants explained 7.7% of the age- and sex-adjusted serum urate variance[3]. The higher proportion of urate variance explained by the replicated CpGs as compared to common genetic variants may partly reflect environmental influences on serum urate levels.

**Causal effects of CpGs on serum urate and gout**. To evaluate whether the urate-associated CpGs might causally affect serum urate levels, we used two-sample MR analysis, where genetic variants associated with DNA methylation of a CpG (methylation quantitative trait locus, meQTL) in *cis* (<500 kb) were used as proxies or genetic instruments of the CpGs. *Cis* meQTLs were available for 27 of the 99 replicated CpGs from a meta-analysis of cohorts of EA in the Genetics of DNA Methylation Consortium (GoDMC, Methods). Combined with summary statistics of a GWAS of serum urate among EA individuals, we found evidence for significant causal effects of DNA methylation on serum urate levels at four CpGs at *SLC2A9* ($p < 1.9E–3 = 0.05/27$) based on the multiplicative random effect inverse-variance weighted method, our primary method (Methods, Table 2). In sensitivity analysis, all significant causal effects were supported by three or more other methods that are robust to pleiotropy (Supplementary Data 10). After additionally removing the meQTLs that were correlated with any of the five GWAS index SNPs at *SLC2A9* among EA persons (Methods), cg13841979 was the only CpG with ≥4 meQTLs for MR analysis and showed nominally significant causal effects on serum urate levels ($p = 2.25E–2$, Supplementary Data 10). This attenuation of causal effect is consistent with the observation that DNA methylation at cg13841979 mediated the effects of two urate GWAS index SNPs on serum urate, as reported below.

As expected, the causal effects of the four CpGs on serum urate were consistent with their effect direction from EWAS. At cg11266682, a promoter-associated CpGs, higher DNA methylation levels were associated with higher serum urate levels (effect size: 0.21 mg/dL per standard deviation [SD] of DNA methylation beta value, $p = 8.8E–04$). In contrast, for three CpGs further downstream of the transcription, higher DNA methylation levels were associated with lower serum urate levels (effect size: −0.65 to −0.46 mg/dL per SD of DNA methylation beta value, $p < 2.1E–4$). At all four CpGs, little evidence of pleiotropy, a threat to MR assumptions, was detected based on the Egger intercept test (all $p > 0.14$). The observed heterogeneity of the meQTL effects on serum urate levels ($p$ heterogeneity <7.54E–239; Table 2 and Supplementary Figs. 3A to 3D) suggests complex genetic

**Table 2 CpGs with significant causal effects on serum urate or gout from Mendelian randomization analysis.**

| Probe ID | Chr | Position (b37) | Nearest gene | No. of meQTLs used for MR analysis | Odds ratio | Effect size[c] | SE | *p* value | *p* value for pleiotropy | Q statistic for heterogeneity | *p* value for heterogeneity |
|---|---|---|---|---|---|---|---|---|---|---|---|
| *Causal effects of CpGs on serum urate* | | | | | | | | | | | |
| cg02387843 | 4 | 9892887 | SLC2A9 | 6 | – | −0.52 | 0.14 | 2.14E−04 | 2.40E−01 | 1109.2 | 7.54E−239 |
| cg13841979[a] | 4 | 9990048 | SLC2A9 | 10 | – | −0.46 | 0.12 | 1.38E−04 | 1.49E−01 | 1836.5 | <1E−300 |
| cg03725404[b] | 4 | 9998017 | SLC2A9 | 8 | – | −0.65 | 0.14 | 4.36E−06 | 1.95E−01 | 1461.0 | 1.53E−312 |
| cg11266682[a,b] | 4 | 10021025 | SLC2A9 | 11 | – | 0.21 | 0.06 | 8.80E−04 | 3.33E−01 | 3173.1 | <1E−300 |
| *Causal effects of CpG on gout* | | | | | | | | | | | |
| cg03725404[b] | 4 | 9998017 | SLC2A9 | 8 | 0.43 | −0.85 | 0.14 | 2.98E−09 | 1.01E−01 | 105.8 | 1.52E−20 |

Odds ratio and log(odds ratio) of CpGs on gout were estimated per SD in rank-based transformed DNA methylation beta levels.
*P* values from MR analysis were two-sided and obtained from inverse-variance multiplicative random effect method.
*P* value for heterogeneity was based on Cochran's Q test.
The cis methylation QTLs (meQTLs) used as instruments for the urate-associated CpGs were provided by GoDMC ($N \leq 27,750$). The summary statistics of serum urate and gout were from individuals of European ancestry ($N = 288,649$ for urate and $N = 692,537$ for gout). In total, 27 of the urate-associated CpGs had ≥4 meQTLs for MR analysis (Methods). The significance level was set at 1.9E−3 (=0.05/27).
*Chr* Chromosome, *MR* Mendelian randomization, *SE* standard error, *SD* standard deviation.
[a]CpGs with significant mediating effect for two urate GWAS index SNPs (Supplementary Data 12 and 13).
[b]CpGs associated with gene expression of *SLC2A9* in monocytes (Supplementary Data 14).
[c]Effect size of CpGs on serum urate estimated in mg/dL per SD in rank-based transformed DNA methylation beta levels.

influences on DNA methylation in this region. Together, these observations at *SLC2A9* are consistent with a causal effect of differential DNA methylation on serum urate levels.

The same 27 CpGs that were tested for causal effects on serum urate were also tested for their causal effects on gout. The CpG cg03725404 at *SLC2A9* showed a significant causal effect on gout ($p < 1.9E–3 = 0.05/27$, Table 2). Consistent with its causal effect on lower serum urate, cg03725404 conferred lower odds for gout, with an odds ratio of 0.43 per SD of DNA methylation beta value ($p = 2.98E–9$). In sensitivity analysis, the causal effect of cg03725404 for gout was supported by three MR methods robust to pleiotropy (Supplementary Data 11). The scatter plots of the effects of the meQTLs on DNA methylation and gout along with the regression slope from the MR methods showed consistency among MR methods (Supplementary Fig. 4). The meQTLs included in the MR analysis for serum urate and gout were independent of each other (Supplementary Figs. 5A–D and 6, respectively). Leave-one-out analysis showed that the significant causal effects of DNA methylation on serum urate and gout were not driven by any single meQTL at any of the CpGs (Supplementary Fig. 7A–D for serum urate and Supplementary Fig. 8 for gout). Forest plots of the effects of the meQTLs showed that the majority of the meQTLs supported the significant causal effects, despite the presence of heterogeneity (Supplementary Fig. 9A–D for serum urate and Supplementary Fig. 10 for gout). Together, these findings show robust results for the performed MR analyses and support a causal effect of differential DNA methylation at *SLC2A9* on serum urate levels and gout risk. The reverse MR analysis that assessed the potential causal effects of serum urate on DNA methylation levels did not detect any significant findings among the 99 tested CpGs (Supplementary Note 2).

**CpGs at SLC2A9 mediate genetic effects**. Given that four CpGs at *SLC2A9* had significant causal effects on serum urate, we hypothesized that these causal effects may mediate genetic effects at this locus. We previously identified four independent intronic or intergenic independent index SNPs in two neighboring 1 Mb regions at *SLC2A9* in a large-scale GWAS meta-analysis of serum urate of EA populations[3]. Therefore, we first tested whether the four CpGs mediated the effects of any of four independent SNPs on serum urate among EA participants of the ARIC study ($n = 637$). Two CpGs (cg11266682 and cg13841979) had significant mediating effects ($p < 0.0125$) for two of the independent index SNPs (rs10017305 and rs6825187; Supplementary Data 12). We next assessed these findings in two additional studies, KORA (Cooperative Health Research in the Region of Augsburg, $n = 1636$) and SHIP (Study of Health in Pomerania, $n = 223$), and observed significant evidence for DNA methylation as a mediator of the effects of these SNPs on serum urate levels. From the meta-analysis of all three studies, the mediating effects ranged from 21% by cg13841979 for rs10017305 (mediating effect: 0.05 mg/dL, $p = 4.7E–07$) to 43% by cg11266682 for rs6825187 (mediating effect: 0.10 mg/dL, $p = 3.0E–5$, Supplementary Data 13).

**Urate-associated CpGs, gene expression, and chromatin accessibility**. An important function of DNA methylation is the regulation of gene expression[25]. To determine whether the urate-associated CpGs were associated with gene expression in blood cells, we used summary statistics of DNA methylation and gene expression from monocytes and whole blood. Gene expression in monocytes is particularly relevant for *SLC2A9*, which is mainly expressed in monocytes and myeloid dendritic cells among the blood cell types (Supplementary Fig. 11)[26]. Of the 99 replicated CpGs, 26 CpGs were significantly associated with gene expression in *cis* in monocytes ($p < 5E–4$; Supplementary Data 14)[27]. Some of the genes harbored multiple urate-associated CpGs (two at *PHGDH*, five at *SLC2A9*, six at *SLC1A5*). At *PHGDH* and

*SLC1A5*, higher DNA methylation levels were associated with lower gene expression, with CpGs mapping to intron 1 or intron 2 of the gene, depending on the reference isoform (Fig. 3). The inverse association between DNA methylation levels in intron 1 and gene expression has been reported to be conserved across vertebrates and is found in multiple human tissues[28]. Interestingly, at *SLC2A9*, the effect direction on gene expression for the five CpGs differed by genomic location (Fig. 4A, B). The three CpGs annotated as promoter-associated and located at introns 1 or 2 of isoform 1 (cg02734326, cg00071950, cg11266682) showed inverse associations with gene expression (e.g., cg11266682: effect −0.037, $p = 4.16E–15$; Fig. 4A and Supplementary Data 14). These three promoter-associated CpGs were associated with higher serum urate levels in EWAS (Fig. 4C). In contrast, cg03725404, located further downstream in the direction of transcription, also showed inverse associations with gene expression (effect −0.024, $p = 4.75E–6$, Fig. 4B) but was associated with lower serum urate levels in EWAS (Fig. 4C). Both cg11266682 and cg03725404 were also shown to have significant causal effects on serum urate, as reported above. The different directions of association across CpGs in *SLC2A9* indicate complex regulation. Assuming that DNA methylation affects gene expression levels, which in turn influence serum urate levels, the observation that higher DNA methylation levels at the three promoter-associated CpGs in *SLC2A9* were associated with lower gene expression in monocytes and higher serum urate levels (Fig. 4B, C) allows for the inference that *SLC2A9* expression in monocytes due to differential methylation at these promoter-associated CpGs has an inverse relationship with serum urate levels. As depicted in Fig. 5A, cg11266682 was associated with lower gene expression levels but showed a positive, causal effect on serum urate levels. In contrast, cg03725404, a CpG further downstream in the direction of transcription, was also associated with lower gene expression but had an inverse, causal effect on serum urate levels (Fig. 5B).

For gene expression in whole blood, eight CpGs showed significant associations (Supplementary Data 15). For all seven CpGs with significant associations with gene expression in both monocytes and whole blood (two in *PHGDH*, one in *PARP3*, two in *UAP1L1*, and two in *SLC1A5*), the directions of effect in the two gene expression datasets were consistent. CpGs at *SLC2A9* were not associated with gene expression in whole blood.

There are two main transcripts of *SLC2A9*. Isoform 1 encodes the long version of the GLUT9 protein, which has been reported to localize to the basolateral membrane of renal proximal tubule cells, where it mediates urate reabsorption[29]. We investigated whether the DNA methylation pattern of the urate-associated CpGs in the kidney may be similar to those observed in whole blood. Whole genome bisulfite sequencing (WGBS) of kidney samples ($n = 5$) showed lower DNA methylation levels at the urate-associated CpGs in the promoter-associated region of isoform 1, and the pattern of DNA methylation level aligned with gene expression levels from RNA-sequencing (Fig. 6A). We further studied whether the urate-associated CpGs mapped into open chromatin regions, where transcription factor binding may occur. ATAC-seq data from primary human kidney tissue ($n = 3$ each from micro-dissected cortex and medulla) showed that the three promoter-associated CpGs, and—to a lesser extent cg20479063—mapped to DNA sequences with high chromatin accessibility in the kidney (Fig. 6B). Taken together, the data from the kidney samples suggest that the urate-associated CpGs at *SLC2A9* are in a regulatory region with potential effects on the expression of *SLC2A9*.

Lastly, we investigated differential gene expression in kidney tissue using two resources. Of the six urate-associated CpGs in *SLC2A9* that were associated with differential *SLC2A9* expression

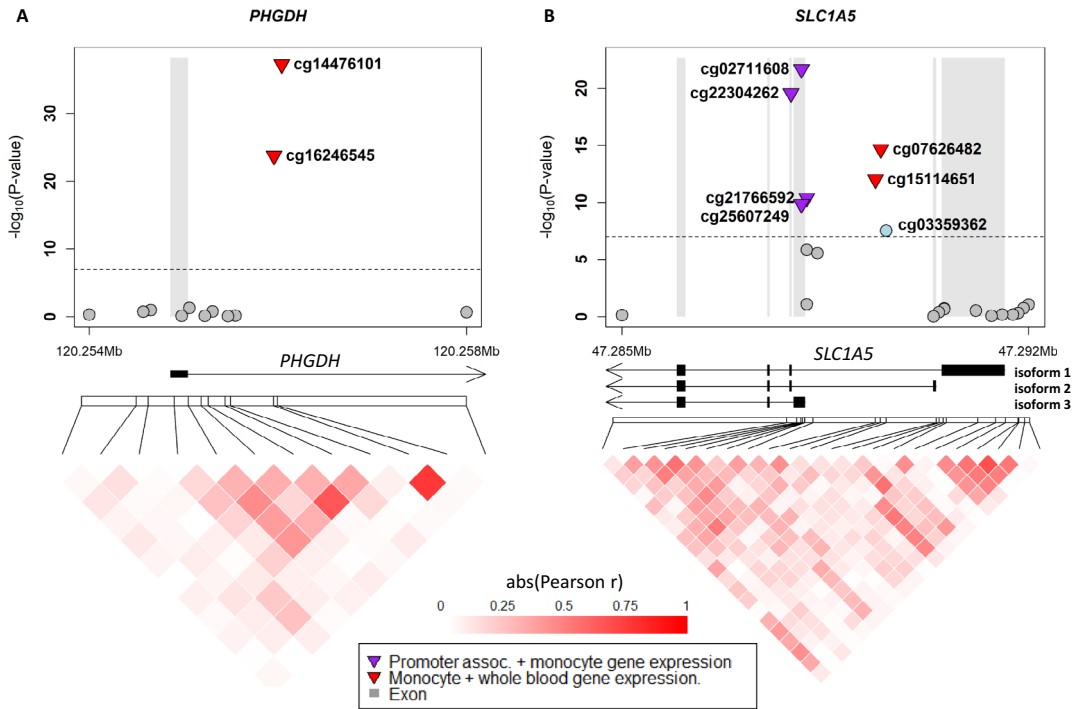

**Fig. 3 Associations of CpGs at *PHGDH* and *SLC1A5* with serum urate levels.** For both *PHGDH* (**A**) and SLC1A5 (**B**), the upper part shows chromosomal positions on the x-axis and the –log10(p value) on the y-axis, and the lower part shows the correlations between DNA methylation levels of the CpGs. The replicated CpGs are labeled. Promoter-associated annotation was based on the HM450K annotation file. The gene models were based on RefSeq curated genes. Associations of DNA methylation with monocyte gene expression were from Kennedy et al. BMC Genomics 2018 and with whole blood from the KORA study (Methods). The correlations between the CpGs were generated using DNA methylation data from 804 European American participants of the ARIC study. Gene models were based on Genbank RefSeq (Accession numbers. *PHGDH*, NM_006623; *SLC1A5* isoform 1, NM_005628; isoform 2, NM_001145144; isoform 3 NM_001145145). assoc. associated.

in monocytes, four did not show significant associations based on DNA methylation and gene expression levels quantified from micro-dissected human tubule tissue, and two did not meet quality control criteria (Methods, Supplementary Data 16). In the future, investigation of specific *SLC2A9* isoforms and/or of specific kidney cell types may shed light on the potential different effects of DNA methylation on gene expression levels in the kidney and monocyte. Second, we evaluated whether the genes implicated by the urate-associated CpGs from our study showed evidence for differential expression in kidney or intestine of a humanized mouse model of hyperuricemia (Methods). Of the genes nearest to the urate-associated CpGs, four genes were differentially expressed at false discovery rate (FDR) < 0.05 (*ANKRD11*, *CEBPB*, *SLC7A11*, and *UAP1L1*). *SLC7A11* had lower expression in the intestine of mice with hyperuricemia ($\log_2$ fold change: −1.34, FDR = 2.9E−2). *ANKRD11* had lower expression in the kidney ($\log_2$ fold change: −0.24, FDR = 5.6E−3; Supplementary Data 17). These findings are consistent with the urate-associated CpGs being downstream of factors influencing serum urate levels in mice.

**Enrichment of urate-associated CpGs in DNase I hypersensitive sites and transporters**. The urate-associated CpGs were enriched in DNase I hypersensitive sites of all blood cells tested (FDR $q < 0.01$), with the strongest enrichment observed in CD3 cells ($q < 1e−9$). Among histone modifications, H3K4me1, often found at active and primed enhancers, and H3K4me3, often found at active promoters, were enriched in all blood cells tested. While the former were most enriched in primary monocytes and primary hematopoietic stem cells, H3K4me3 showed the strongest enrichment in primary hematopoietic stem cells (Supplementary Figs. 12 and

13)[30,31]. In contrast, H3K9me3 and H3K27me3, generally associated with heterochromatin[30–33], were not significantly enriched in any blood cells or tissues tested (Supplementary Figs. 14 and 15). In addition, among transcription factor binding sites, the urate-associated CpGs were most enriched in POLR2A, the largest sub-unit of RNA polymerase II, the major enzyme synthesizing mRNA in eukaryotes ($p < 1E−11$, Supplementary Fig. 16). Together, the observed enrichments suggest a role of the urate-associated CpGs in transcriptional regulation.

We also assessed the enrichment of genes implicated by urate-associated CpGs in Gene Ontology (GO) terms, and the Kyoto Encyclopedia of Genes and Genomes (KEGG) and Reactome pathways (Methods). After correction for multiple testing, significant enrichment was observed for 55 terms (Supplementary Data 18). The terms or pathways with the strongest enrichment were related to transmembrane transport of organic acids, carboxylic acids, amino acids, organic and inorganic anions, as well as processes in leukocytes and myeloid cells (Fig. 7). These terms suggest that differentially methylated CpGs capture information about small molecule transport in urate homeostasis.

**Urate-associated CpG sites and other traits**. Our prior work on the genetics of serum urate revealed significant genetic correlations between serum urate and many cardiometabolic traits[3]. Using published EWAS summary statistics, we investigated whether these relationships were also observed when studying epigenetic variation by examining the association between replicated urate-associated CpGs and other cardiometabolic traits (Methods, Supplementary Data 19). Liver and kidney traits were also included due to the important role of these two organs in urate production and excretion[34]. Most of these traits were

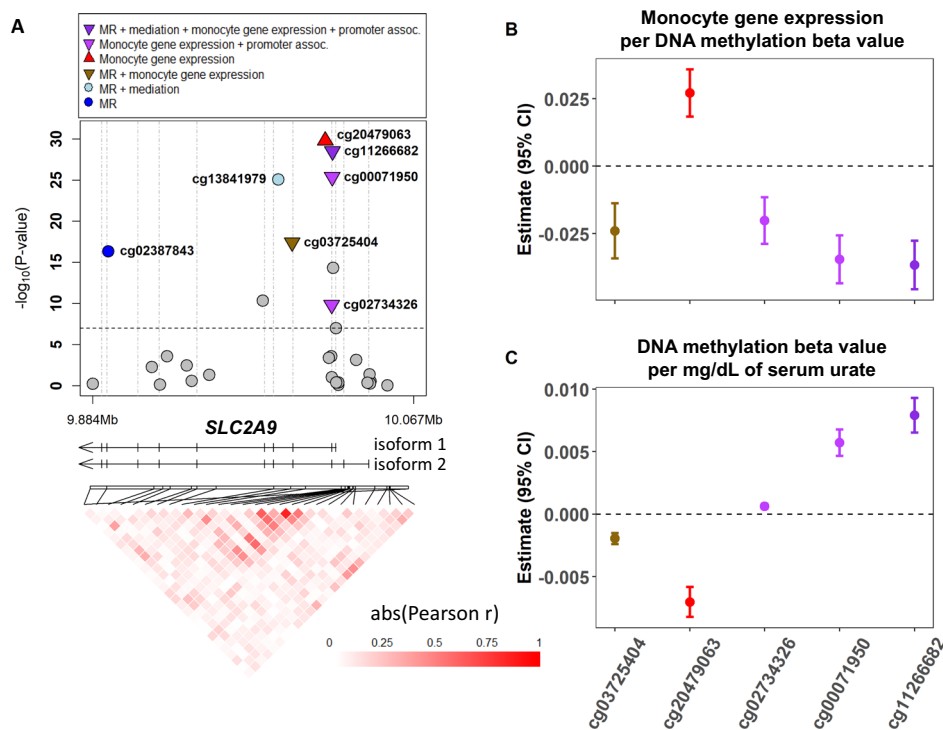

**Fig. 4 Associations of CpGs at *SLC2A9* with serum urate levels and gene expression in monocytes.** CpGs in the *SLC2A9* region (**A**) with the effect size of DNA methylation on gene expression in monocytes at five replicated CpGs (**B**) and the effect size of serum urate on DNA methylation levels at these same five CpGs (**C**). All labeled CpGs were replicated. In the legend, MR indicates CpGs with a significant causal effect on serum urate levels based on Mendelian randomization analysis. Colors of the estimates in panels **B** and **C** match the color legend in panel **A**. Promoter-associated annotation was based on the Illumina HM450K annotation file. The gene models were based on RefSeq curated genes. The correlations between the CpGs were generated using DNA methylation data from 804 European Americans participants of the ARIC study. The association between DNA methylation and monocyte gene expression in panel **B** is based on Kennedy et al. BMC Genomics 2018 (*n* = 1202). The DNA methylation estimates in panel **C** are based on the meta-analysis combining discovery and replication cohorts in the present study (*n* = 17,996). Genbank RefSeq accession numbers: isoform 1 (NM_020041), isoform 2 (NM_001001290). MR Mendelian randomization, assoc. associated.

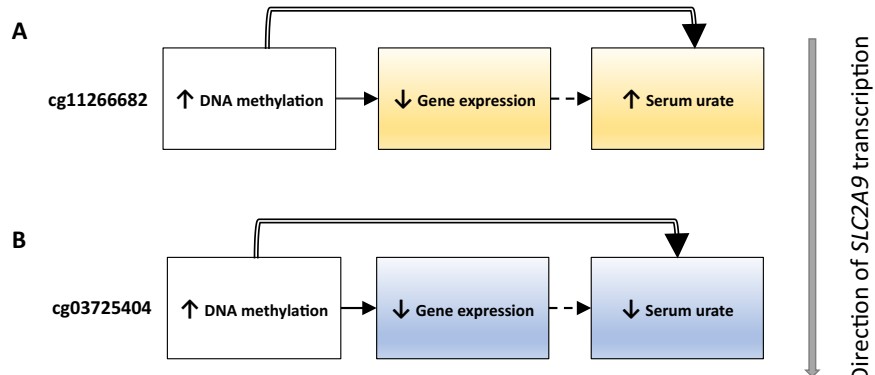

**Fig. 5 Conceptual figure summarizing patterns of relationships between DNA methylation at *SLC2A9*, its expression in monocytes, and serum urate levels.** The figure focuses on the two CpGs with significant causal effects on serum urate and association with gene expression in monocytes. The inferred relationship between gene expression and serum urate levels is inverse for the promoter-associated CpG cg11266682 (**A**, orange shading), and concordant for CpG cg03725404 (**B**, blue shading). Solid compound arrows indicate both observed cross-sectional association and causal effect of DNA methylation on serum urate based on Mendelian randomization analysis. Solid arrows indicate observed cross-sectional associations. Dashed arrows indicate inferred relationships.

included as covariates in the EWAS of serum urate with the goal of detecting urate-specific DNA methylation signals (Methods). Of the replicated urate-associated CpGs, 17 had significant associations with at least one cardiometabolic trait (Fig. 8 and Supplementary Data 20). Interestingly, the directions of association were always consistent with the correlation between serum urate and these traits in epidemiological studies (Fig. 8)[7,35–37]. A

few urate-associated CpGs showed associations with a particularly high number of cardiometabolic traits: cg14476101 annotated to *PHGDH* (six traits), cg06690548 annotated to *SLC7A11* (seven traits), and cg02711608 annotated to *SLC1A5* (six traits). Lower DNA methylation level were associated with higher serum urate levels, as were as higher levels of body mass index (BMI), blood pressure, triglycerides, liver enzymes, C-reactive protein

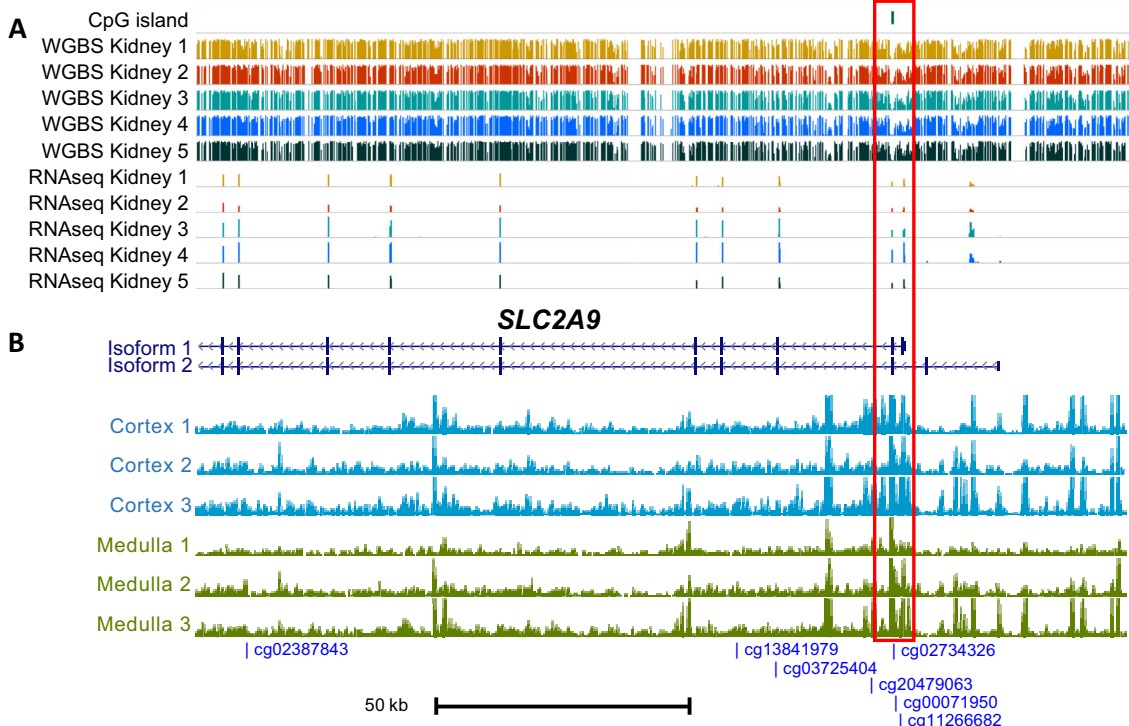

**Fig. 6 Whole genome bisulfite sequencing and ATAC-sequencing data of kidney tissue at SLC2A9.** Whole genome bisulfite sequencing of kidney samples (n = 5) showed that lower DNA methylation levels at CpGs localized to the promoter-associated region in isoform 1 on *SLC2A9* and aligned with gene expression levels from RNA-sequencing (**A**). ATAC-sequencing data from primary human kidney tissue (n = 3 from micro-dissected cortex and medulla each) showed that the three promoter-associated CpGs, and—to a lesser extent cg20479063—mapped to DNA sequences with high chromatin accessibility in the kidney (**B**). Gene models were based on Genbank RefSeq (accession numbers: isoform 1, NM_020041; isoform 2, NM_001001290).

(CRP), and incident diabetes in their respective EWAS (Fig. 8). These traits represent components of the metabolic syndrome or are strongly correlated with the components[38], as is serum urate[37]. Given that the EWAS of the cardiometabolic traits adjusted for white blood cell composition in their analysis, the associations between DNA methylation and traits at the CpGs are independent of cell type composition. Together, these associations at these CpGs may reflect DNA methylation signatures of metabolic syndrome in blood. In contrast, none of the 99 replicated urate-associated CpGs were associated with fasting glucose and insulin among individuals without diabetes, who are less likely to be affected by the metabolic syndrome. The urate-associated CpGs were also not found among those associated with high-density and low-density cholesterol (HDL-C, LDL-C) and estimated glomerular filtration rate (eGFR). In addition, 4 of the 17 CpGs that were associated with serum urate and cardiometabolic traits (cg16246545 at *PHGDH*, cg19693031 at *NBPF20/TXNIP*, cg06690548 at *SLC7A11*, and cg06690548 at *CPT1A*) were associated with mostly lipid-related metabolites, quantified using nuclear magnetic resonance (Supplementary Data 21)[39]. The associations of these four CpGs were largely observed with triglycerides in various lipid subfractions or lipid metabolites that are part of very low-density lipoprotein and had concordant effect directions as their associations with serum urate.

## Discussion

This large-scale EWAS, based upon data from up to 17,996 participants largely drawn from population-based studies, identified and replicated 99 CpGs, at which differential DNA methylation was significantly associated with serum urate levels. The genes implicated by these CpGs were strongly enriched in terms and/or pathways related to small molecule transport, including organic anions and amino acids. MR analyses supported significant causal effects of some CpGs on serum urate levels and gout at *SLC2A9*, the strongest GWAS locus for serum urate that encodes a major urate transport protein in the kidney. Mediation analyses supported that genetic variants at *SLC2A9* affect serum urate levels partly via epigenetic mechanisms. Moreover, 17 urate-associated CpGs showed significant associations with numerous cardiometabolic traits, with the direction of association always consistent with the ones between serum urate and the respective cardiometabolic trait reported from observational studies. Differential DNA methylation at these sites may therefore reflect an epigenetic signature of cardiometabolic traits in whole blood.

Results from dedicated EWAS of serum urate levels have not been reported previously. One previous EWAS of blood metabolite levels in one of our contributing studies, KORA F4, examined urate as one of 649 metabolites, and reported significant associations between DNA methylation at cg00071950 in *SLC2A9* and serum urate levels[40]. This EWAS meta-analysis of measured serum urate levels lends further support to the reproducibility and validity of this prior finding through successful replication in different study populations, association with gene expression, and biological plausibility.

In addition to replicated associations between DNA methylation and serum urate, our study reveals significant causal effects of DNA methylation at some CpGs of *SLC2A9* on serum urate and gout. Prior exposure to serum urate in its soluble or crystal form has been shown to heighten the proinflammatory response of myeloid cells in vitro and in animal models potentially through epigenetic mechanisms[41]. This is also known as urate-induced training immunity. In our study, MR analysis did not identify significant causal effects of serum urate on DNA methylation, but

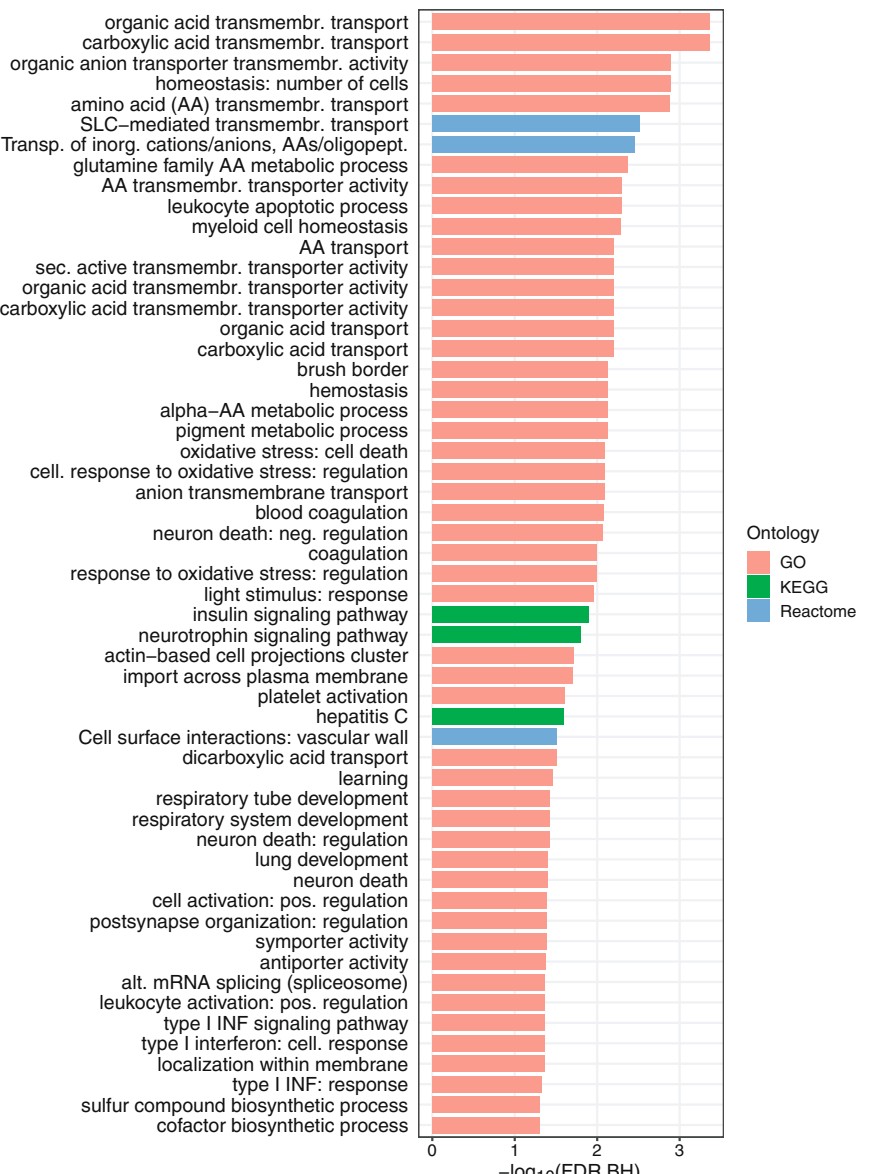

**Fig. 7 GO terms and KEGG and Reactome pathways that were enriched for urate-associated CpGs.** Significance level was set at false discovery rate <0.05. GO Gene Ontology, KEGG Kyoto Encyclopedia of Genes and Genomes.

it is conceivable that serum urate might act on other forms of epigenetic mechanisms, such as histone modification[42].

Serum urate levels show high genetic and observational correlations with kidney function measures and with cardiometabolic traits. Previous large EWAS of kidney function measures focused on eGFR in population-based studies[43] or on changes in kidney function among patients with kidney disease[44]. Despite the strong, inverse genetic and observational correlations between serum urate and eGFR and the well-established role of the kidney in serum urate homeostasis, the replicated urate-associated CpGs in our study did not show significant associations when assessed in EWAS of eGFR. In contrast, many urate-associated CpGs showed significant associations in results from EWAS of cardiometabolic traits, including triglycerides[45], blood pressure[46], BMI[47], liver enzymes[48], CRP[49], as well as diabetes[50]. Many of these traits are either conditions that define the metabolic syndrome, or directly related to such conditions (high triglyceride levels, high blood pressure, high BMI, high blood glucose levels)[38]. There has been no robust evidence supporting causal effects of serum urate on cardiometabolic traits[51,52]. Instead, our observations are consistent with shared gene regulatory programs resulting in a common DNA methylation signature of serum urate and metabolic syndrome in whole blood.

Our study also provides additional insights into a major transporter of serum urate. *SLC2A9* has consistently been detected as the strongest locus in GWAS of serum urate with multiple independent signals[1,3,16,53–57], but the mechanism underlying the GWAS signal is largely unknown. The association between genetic variants at *SLC2A9* and serum urate levels may be mediated by transcript- or tissue-specific regulatory mechanisms. *SLC2A9* has two described isoforms whose gene products have equivalent urate transport activity but differ in the N-terminal cytosolic portion of the protein[58]. *SLC2A9* shows high complexity in human kidney tissue; the two isoforms of *SLC2A9* have been reported to differ in cellular localization possibly affecting their function in urate homeostasis. Isoform 1 was found to be expressed at the basolateral side of the proximal tubule, while isoform 2 was expressed at the apical side of the collecting duct[29]. The functional consequences of this complexity are not yet clear. In our study, we observed complex relations of DNA

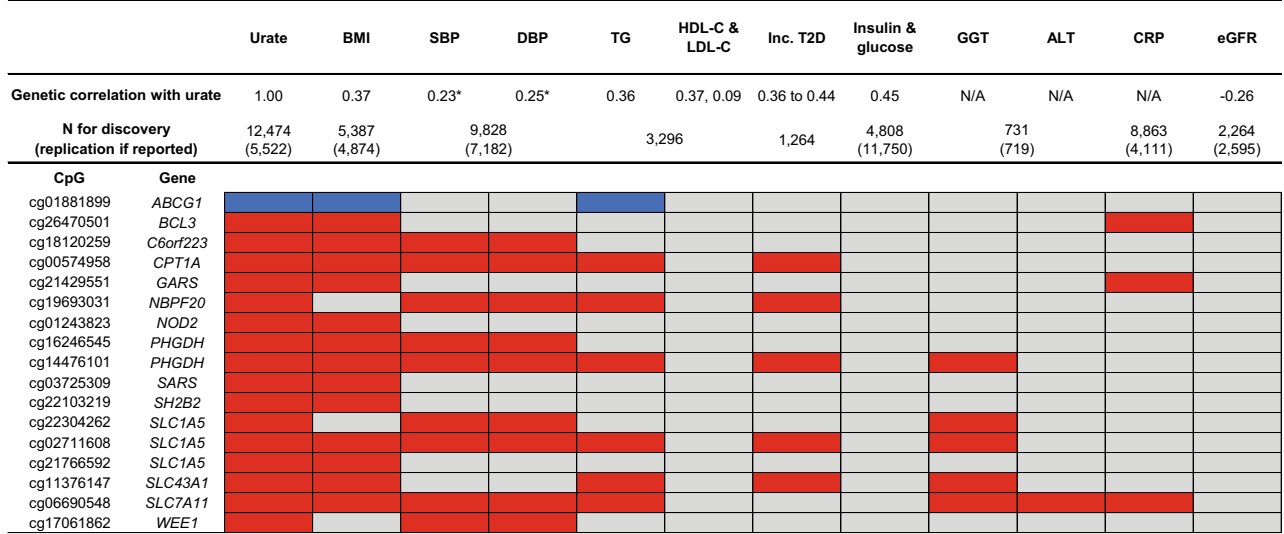

**Fig. 8 Urate-associated CpGs that were associated with cardiometabolic and kidney traits.** All traits that were included in the lookup are shown, including those without reported associations with urate-associated CpGs. Blue: positive association between DNA methylation and trait levels; Red: inverse association between DNA methylation and trait levels. BMI body mass index, SBP systolic blood pressure, DBP diastolic blood pressure, TG triglycerides, HDL-C high-density lipoprotein cholesterol, LDL-C low-density lipoprotein cholesterol, inc. incident, GGT Gamma-glutamyl transferase, ALT alanine aminotransferase, CRP C-reactive protein, eGFR estimate glomerular filtration rate. Genetic correlations were reported in Tin et al. Nat Genet 2019. *Genetic correlation between serum urate and hypertension was 0.39. No EWAS of hypertension was available; SBP and DBP were used instead.

methylation levels at *SLC2A9* with gene expression in monocytes and serum urate. Our findings support an inferred inverse relationship between DNA methylation-related *SLC2A9* expression in monocytes and serum urate levels at some CpGs. An inverse relationship between GLUT9 expression in the kidney and serum urate would be difficult to explain. Loss-of-function mutations at *SLC2A9* in humans result in type 2 familial renal hypouricemia (OMIM# 612076) supporting a role for GLUT9 as a high-capacity urate transporter in tubular urate reuptake from urine into blood. Therefore, a positive association between *SLC2A9* gene expression in the kidney and serum urate levels would be expected[58].

However, *SLC2A9* is expressed in a range of extra-renal tissues, with high abundance in the liver and lower abundance in the heart, lung, and intestine[59–61]. In mouse hepatocytes, GLUT9 is expressed in the basolateral membrane[60], across which urate uptake from blood into the cell can occur[62]. Therefore, the role of GLUT9-mediated transport of urate may potentially affect serum urate levels in different directions, depending on transcript isoform and tissue localization. For example, loss of function mutations in dogs and liver-specific inactivation of *SLC2A9* in a mouse model result in hyperuricemia[62,63]. In addition, basolateral localization of GLUT9 has also been reported in mouse enterocytes of the small intestine[64], and enterocyte-specific *Slc2a9* knock-out resulted in reduced intestinal urate excretion and hyperuricemia[65]. A role for *SLC2A9* in intestinal excretion would also be consistent with an inverse relationship between DNA methylation-related *SLC2A9* expression and serum urate levels. Therefore, it is conceivable that the DNA methylation signals we detected at *SLC2A9* in blood represent a signature for urate transport into enterocytes, hepatocytes, and/or other cell types, as a constituent of a urate secretion pathway, a mechanism that would lower serum urate levels. Roles for *SLC2A9* in urate uptake into cells in both secretory and reabsorption pathways may also help explain some of the heterogeneity of associations of SNPs in the *SLC2A9* locus[3]. Regardless of the mechanism, the finding that urate-associated genetic variants at *SLC2A9* could affect serum urate levels via epigenetic mechanisms may be of potential clinical relevance, as each SD higher DNA methylation at CpGs at *SLC2A9* conferred up to 57% lower odds of gout, the most common form of inflammatory arthritis in adults.

Terms or pathways enriched for urate-associated CpGs were mainly related to transmembrane transport of organic acids, carboxylic acids, amino acids, and organic and inorganic anions. This does not only align with uric acid as an organic acid, but also with several genes encoding enzymes and transporters related to amino acid generation and movement across membranes. For example, *PHGDH* encodes 3-phosphoglycerate dehydrogenase, which catalyzes the oxidation of 3-phosphoglycerate to 3-phosphohydroxypyruvate, the committed step in the biosynthesis of L-serine. Inhibition of PHGDH alters nucleotide metabolism via disruption of mass balance within central carbon metabolism, leading to changes in the pentose phosphate pathway and the tri-carboxylic acid cycle[66]. In addition, *PHGDH* also has a role in lipid metabolism. Knockdown of *PHGDH* in liver cell lines resulted in lower expression of *LPL*, the gene encoding lipoprotein lipase, and higher expression of *LDLR* and *ABCA1*, both important in lipid transport[67]. The genes *SLC1A5*, *SLC7A11*, and *SLC43A1* encode transport proteins for neutral amino acids[68], cysteine as well as glutamate[69], and for branched-chain amino acids[70], respectively. The CpGs in genes involved in the transport of small molecules such as amino acids, represented the CpGs with the strongest associations in this EWAS of serum urate. These CpGs were also associated with other cardiometabolic traits. Taken together, these results suggest that shared epigenetic regulatory mechanisms may have an important role in urate homeostasis[15,71–73].

Strengths of our study include its large sample size, the replication of findings in separate study samples, and the use of rigorous statistical methods. The agreement of findings based on multiple layers of complementary evidence and the biological plausibility of many of the results support the validity of our findings. These findings provide new insights into the epigenetic regulation of serum urate levels that are complementary to genetic association studies of serum urate. Potential limitations of our study relate to the quantification of DNA methylation from whole blood or from specific blood cell populations, which may not be representative of other important organs of urate homeostasis such as the kidney or liver. However, serum urate was also measured in blood, where its levels may be influenced by metabolic functions beyond simply reflecting purine metabolism. In addition, serum urate concentrations play an important role in gout,

making blood an attractive target tissue for studies of urate homeostasis and gout.

In summary, meta-analyses of EWAS of serum urate levels and downstream characterization identify urate-associated CpGs that explain a substantial proportion of serum urate variance, mediate some of the genetic effects on serum urate levels, and show associations with several cardiometabolic traits consistent with the observed relationships between serum urate and these traits. These findings constitute complementary evidence to insights from GWAS of serum urate levels.

## Methods

**Study samples and workflow.** The discovery and replication analyses were conducted based on a pre-specified analysis plan. A script (https://github.com/genepi-freiburg/ckdgen-pheno/tree/ckdgen-ewas-pheno) was circulated to all participating cohorts (Supplementary Data 1) to generate summary files that were inspected by a central analysis team. All studies were community-based or non-clinical populations. Each cohort conducted study-specific EWAS and uploaded epigenome-wide summary statistics. Before the meta-analysis, we conducted quality control of study-specific results, which included the assessment of the distributions of effect sizes, standard errors, and p values within each study and across studies. Supplementary Data 2 reports study-specific methods, including the DNA methylation detection p value and processing pipeline. Supplementary Data 3 reports the white blood cell proportions in each cohort, which were used as covariates to estimate associations between urate and DNA methylation independent of cell type composition. Study research protocols were approved by the respective ethics committees. All participants in all studies provided written informed consent.

**DNA methylation quantification and quality control.** Genomic DNA was extracted from peripheral blood. Levels of DNA methylation were quantified using the Infinium MethylationEPIC BeadChip array (EPIC, six studies), the Illumina Infinium HumanMethylation450K BeadChip array (HM450K, 17 studies), or the Illumina Infinium HumanMethylation27 BeadChip array (HM27K, one study). DNA methylation data preprocessing was performed according to individual study protocols, which included background correction, quantile normalization, probe filtering, sample filtering, SNP matching to the SNP control probe locations, outlier filtering, and assay type correction (Supplementary Data 2).

**Study-specific EWAS.** Ancestry-specific association analyses were conducted within each study. Among the 24 contributing cohorts, 14 cohorts excluded participants who were on urate-lowering medications, whereas the other cohorts did not have this information available (Supplementary Data 1). The methylation levels at each CpG probe were represented as beta values, which can be interpreted as the proportion of CpG sites that were methylated. The beta values were analyzed as the dependent variable with serum urate as the independent variable in a linear regression model adjusting for age, sex, BMI, current smoking, eGFR, HDL-C, systolic blood pressure (SBP), log-transformed levels of CRP and triglycerides, genetic principal components (PC), and estimated or measured blood cell type proportions in order to allow for detecting associations between serum urate and DNA methylation levels independent of differences in cell type composition[74]. Additional study-specific technical covariates included control probe PCs[5], study center, processing batch, sentrix ID, and sentrix position, as applicable.

**Discovery and replication meta-analysis.** Studies were separated into discovery and replication cohorts by chronological order of contribution to this project (Supplementary Data 1). Only CpG sites common to both the EPIC and HM450K were included in the meta-analysis ($n = 441,854$). CpG probes overlapping with SNPs were annotated using information from Illumina.

Prior to meta-analysis, each set of study-specific results was adjusted for test statistic inflation using the BACON method, which was developed to control for bias and inflation in EWAS using an empirical null distribution approach and assumes that the test statistics are a mixture of three distributions: negative, positive and null associations[75]. The inflation factor is estimated from the distribution of null association. We conducted inverse-variance weighted fixed effects meta-analysis using the R package metafor (version 2.1-0) for discovery, replication, and for combining the results of discovery and replication[76]. CpG sites were excluded if their available sample size was less than half of the sample size in the respective meta-analyses, or if their $I^2$ heterogeneity estimate was >95%.

The statistical significance threshold for the discovery step was $p < 1.1E–7$ (=0.05/441,854). The replication criteria were: $P_{discovery} < 1.1E–7$, $P_{replication} < 0.05$ with consistent effect direction, and $P_{combined} < 1.1E–7$. For the replicated CpGs, we quantified study heterogeneity within an ancestry using the $I^2$ from a meta-analysis including all studies within an ancestry, namely EA (17 studies) and AA (5 studies). We further quantified heterogeneity among the four ancestry groups using the $I^2$ from a transethnic meta-analysis that combined the summary statistics of the meta-analyses of EA, and AA with those from the LOLIPOP study (SA), and the RODAM study (SSA). Given that there was only a single study each including

participants with SA and SSA ancestry, the $I^2$ from this transethnic meta-analysis might represent both ancestry and study heterogeneities.

**Heritability of replicated CpG sites and explained urate variance.** We used three sources of DNA methylation heritability estimates from whole blood to characterize the replicated urate-associated CpGs: (a) a twin study of 1464 individuals from Great Britain [Hannon et al.], (b) a twin study of 2386 individuals from the Netherlands [van Dongen et al.], and (c) a family-based study of 614 individuals from Australia [McRae et al.][22–24]. Hannon et al. included 426 monozygotic (MZ) and 306 same-sex dizygotic (DZ) twin pairs to estimate the proportion of the variance of DNA methylation explained by additive genetic (A), shared environment (C), and unshared environmental (E) factors[22]. Van Dongen et al. included 769 MZ and 424 DZ twin pairs for the ACE model analysis[23]. McRae et al. studied 614 individuals from 117 families and used intraclass correlation analysis to partition the variance of DNA methylation into additive genetics and environmental components[24]. DNA methylation levels of these three studies were quantified using the HM450K array.

The proportion of serum urate variance explained was calculated based on data from 1832 participants of the KORA-FF4 study with DNA methylation levels quantified using blood samples collected in 2014. Among the participants of the KORA-FF4 study, 988 were also participants of the KORA F4 cohort, one of the discovery cohorts, with DNA methylation levels quantified using blood samples collected in 2007, 7 years earlier than those in the KORA-FF4 study. Data were available for 97 of the 99 replicated, urate-associated CpGs (missing observations for cg08257009 and cg05201185). To enable comparison of the proportion of explained urate variance to the one obtained from age- and sex-adjusted GWAS summary statistics[3], two models were computed: a base model of the proportion of urate variance explained by age and sex, and an extended model that additionally included residuals from a regression of the CpGs on sex, age, and cell type composition. The proportion of serum urate variance explained by the replicated CpGs was then calculated as the difference in coefficient of determination ($R^2$) between the two models.

**Forward Mendelian randomization (MR) analysis: causal effects of DNA methylation on serum urate and gout.** To evaluate whether the urate-associated CpGs might have a causal role in influencing serum urate levels or gout, we conducted two-sample MR analyses using meQTLs as genetic instruments of the replicated CpGs. The random assignment of alleles during meiosis mimics treatment allocation in randomized controlled trials. Using meQTLs as proxies or genetic instruments of DNA methylation enables inferences on the causal effects of DNA methylation on serum urate or gout, provided that the meQTLs meet the instrumental variable assumptions: (1) the meQTLs are associated with DNA methylation, (2) the meQTLs are not associated with potential confounders of the association between DNA methylation and serum urate or gout, and (3) the meQTLs are only associated with serum urate or gout through DNA methylation, i.e., without pleiotropic effects[77]. While assumptions 2 and 3 cannot be fully verified, we have addressed these assumptions in our selection criteria of meQTLs and analysis methods as detailed below.

The meQTLs of the CpGs were selected two steps. In the first step, we addressed the three instrumental variable assumptions. To address assumption 1, we selected independent cis meQTLs as genetic instruments with strong association with DNA methylation ($p < 5E–8$). To address assumptions 2 and 3, we removed those with genome-wide significant association ($p < 5E–8$) with potential confounders, and with stronger associations with the outcome than with DNA methylation, which suggests pleiotropic effects of a meQTL or may lead to reverse causation. In this first step of instrument selection, we selected cis meQTLs for each CpG (within 500 kb of the CpG) from the meta-analysis of the GoDMC with MAF > 1% and $p < 5e–8$ to avoid weak instrument bias[78]. The meQTLs were further pruned based on the following criteria: (a) a clumping threshold of $r^2 < 0.05$, (b) associated with potential confounders (BMI, current smoking, HDL-C, SBP, CRP, and triglycerides) at $p < 5E–8$, (c) failed Steiger filtering, i.e., displayed stronger association with the outcome than the exposure, and (d) failed harmonization checks of the harmonise_data function in the TwoSampleMR package, which matches the effect allele of the SNPs associated with the exposure and the outcome and flags palindromic SNPs with allele frequency close to 0.5 as ambiguous[79,80]. The sources of the GWAS summary statistics of the potential confounders are reported in Supplementary Data 22. In the second step, to ensure that the meQTLs for each CpG were indeed conditionally independent, we conducted conditional analysis using Genome-wide Complex Trait Analysis (GCTA) to obtain the conditional $p$ value for each meQTL controlling for other meQTLs of the same CpG (command: cojo-cond)[81]. As the reference panel, we used 13,558 randomly selected individuals of British descent, and 16,969,363 SNPs with MAF > 0.01% after quality control as reported previously[3]. We retained only meQTLs with conditional $p < 5E–8$ as genetic instruments of the CpGs. The median numbers of meQTLs among the 92 replicated CpGs outside of *SLC2A9* was 2 (25th, 75th percentile: 0, 5), whereas it was 20 (25th, 75th percentile: 8, 21) among the 7 CpGs at *SLC2A9*.

The genetic association summary statistics for meQTLs, serum urate, and gout were generated among EA individuals. The meQTL summary statistics were obtained from a meta-analysis of the GoDMC Consortium (max. $n = 27,750$ from 36 EA studies), which included some of the participating studies of the EWAS of serum urate[82]. DNA methylation in GoDMC was measured in whole blood or cord

blood using the HM450K or EPIC BeadChips. The 1000 Genomes reference panel was used for genotype imputation. The analysis in each cohort had two steps. First, residuals of DNA methylation beta values were generated by adjusting out age, sex, predicted white blood cell type proportions, predicted smoking status based on published results of smoking-associated DNA methylation, and genetic PCs. Next, these residuals were rank transformed and standardized and used for genetic association analysis with adjustment for relatedness in family-based studies. The SNP effect sizes can be interpreted as SD higher DNA methylation per allele. The meta-analysis was conducted in two phases. First, each GoDMC study analyzed the association between all SNPs and all CpGs, returning only associations with $p < 1E–5$. Next, these associations with $p < 1e–5$ were combined to create a candidate list of meQTL associations, which included *cis* associations found in at least one dataset and *trans* associations in at least two datasets. The association statistics of these candidate meQTLs ($n = 120,212,413$) were obtained from all cohorts, and then combined using fixed-effect meta-analyses[82].

The GWAS summary statistic for serum urate ($n = 288,649$) and gout ($n = 692,537$) were obtained from recent GWAS meta-analyses of EA, samples from the CKDGen Consortium[3]. We required each CpG to have at least four cis (<500 kb) meQTLs as genetic instruments. As the primary MR analysis method, we used the multiplicative random effect inverse-variance weighted method as recommended by the guidelines for MR investigation[77]. We used the Egger intercept to evaluate potential pleiotropy, and Cochran's $Q$ test for assessing heterogeneity[78,83,84]. Sensitivity analysis using other MR methods that are robust to pleiotropy were conducted: MR Egger, simple mode, weighted median, and weighted mode[85–87]. To assess whether a causal estimate may be largely driven by a single genetic instrument, we conducted leave-one-out analysis. To assess whether the causal effects of the CpGs at *SLC2A9* on serum urate were partly due to GWAS signals at this locus, we conducted an additional sensitivity analysis excluding meQTLs with $r^2 > 0.05$ with any of five GWAS index SNPs at *SLC2A9* (rs4447862, rs10017305, rs6825187, rs62286563, and rs73224492) among individuals of EA, reported in Tin et al.[3]. The SNP rs4447862 had the lowest $p$ value in the *SLC2A9* region, and the other four SNPs were detected as genome-wide significant independent SNPs by the GCTA stepwise model selection command (cojo-slct)[88]. All MR analyses were conducted using the TwoSampleMR package[79]. Calculations of the minimum detectable effect size in the forward MR analysis of DNA methylation on serum urate or gout supported that the MR analysis was well powered (Supplementary Note 3).

**Reverse MR analysis: causal effect of serum urate on DNA methylation**. To evaluate the potential causal effects of serum urate on DNA methylation levels, we conducted two-sample reverse MR analysis. The genetic instruments for serum urate were 123 index SNPs from a recent GWAS meta-analysis of 288,649 EA participants[3]. The associations between these SNPs and DNA methylation levels were generated among 3866 EA participants of the Framingham Heart Study, a sample independent of the EWAS of serum urate[89]. The ARIC study was not included in the reverse MR because most meQTL data were from AA participants, whereas the urate summary statistics were based on data from EA individuals. The DNA methylation levels were quantified as beta values from the HM450K array. The analysis of the association between SNPs and DNA methylation levels adjusted for age, sex, the top 50 methylation PCs, predicted blood cell fractions, and used linear mixed model to account for family structure. The reverse MR did not use the GoDMC meQTL results employed in the forward MR analysis, because the GoDMC meQTL results only contained SNPs associated with DNA methylation levels below a certain significance level, as reported above. Given that the urate index SNPs were selected in separate 1-Mb regions across the genome, we did not conduct GCTA conditional analysis to reassess the independence among these index SNPs. Otherwise, the selection of the SNPs for serum urate, their harmonization, and MR methods for primary and sensitivity analysis were the same as those in the forward MR analysis.

**Mediation analysis**. Given that four CpGs at *SLC2A9* showed significant causal effects for serum urate, we evaluated whether these CpGs may mediate the genetic effects of the EA index SNPs of serum urate at the *SLC2A9* locus[3]. As reported previously, four independent SNPs in two neighboring 1 Mb intervals in the *SLC2A9* region were identified based on GCTA stepwise model selection[3,81,88]. Mediation analyses of the four CpGs for the four index SNPs were first conducted among 637 EA ARIC participants, controlling for age, sex, study center, and ten genetic PCs using the mediation package version 4.5 in R[90]. This mediation method uses simulation to estimate the average causal mediation effect in a potential outcome framework. The significance threshold was set at 0.0125 (0.05/4 independent index SNPs). For the significant index SNP-CpG pairs, mediation analyses were performed in the SHIP-Trend ($n = 223$) and KORA ($n = 1636$) cohorts. The mediation effects of each CpG were then combined using fixed-effect inverse-variance meta-analysis as implemented in metafor[76].

**Association with gene expression and chromatin accessibility**. We investigated the association of the urate-associated CpGs with gene expression in *cis* in monocytes and whole blood. Monocyte data were obtained from Kennedy et al., who assessed these associations among 1202 individuals from the Multi-Ethnic Study of Atherosclerosis[27].

DNA methylation levels were quantified using the HM450K array, and gene expression levels were quantified using Illumina HumanHT-12 version 3.0 and 4.0 Expression BeadChips. The regression analysis used log-transformed expression levels as the dependent variable and DNA methylation beta values as the independent variable controlling for age, race, sex, and study site.

The association of DNA methylation levels with gene expression in whole blood was obtained from 713 participants of the KORA study. The DNA methylation levels were quantified using the HM450K array, and gene expression levels were quantified using Illumina HumanHT-12 v3 Expression BeadChips. The $\log_2$-transformed gene expression values were regressed on the DNA methylation beta values adjusted for sex and age. Prior to the analysis the technical factors as well as the blood cell type proportions were regressed out of the mRNA and DNA methylation levels, and their residuals were included in the final association model. Statistical significance for the association between urate-associated CpGs and gene expression was defined as $p < 5E–4$ ($=0.05/99$).

To evaluate whether urate-associated CpGs at *SLC2A9* might be associated with gene expression in the kidney, we selected five CpGs at the *SLC2A9* locus that were significantly associated with *SLC2A9* gene expression in monocytes (cg03725404, cg20479063, cg02734326, cg00071950, cg11266682). The DNA methylation and gene expression levels were quantified from tubulo-interstitial kidney tissue of 314 persons. The analysis used gene expression (transcripts per million) as the dependent variable and DNA methylation beta values as the independent variable controlling for age, sex, five genetic PCs, prevalent diabetes and hypertension, BMI, batch factors, bisulfite conversion rate, RNA-sequencing batch, and RNA integrity number.

To explore the epigenetic and gene expression landscape in the *SLC2A9* locus in kidney tissues, we looked up the DNA methylation profiles detected by WGBS and gene expression profiles from RNA-sequencing in five human normal kidneys (GEO accession GSE115098)[91]. The methylation beta values from WGBS and read counts from RNA-sequencing at the *SLC2A9* locus (GRCh37/hg19, chr4:9884000-10067000) were converted to bigwig format. The bigwig files were imported into IGV genome browser for visualization. In addition, we investigated whether the urate-associated CpGs at *SLC2A9* might be mapped into open chromatin regions in the kidney using data generated from ATAC-seq (Assay for Transposase-Accessible Chromatin using sequencing). Kidney cortex and medulla tissues were macro-dissected from uninvolved portions of tumor nephrectomy specimens and snap-frozen. ATAC-seq was performed by ActiveMotif (Carlsbad, California) and the subsequent data were mapped to GRCh38 and normalized by read depth. Peak calling was performed using MACS 2.1.0 and filtered against the ENCODE blacklist. Master list generation and set operations with the resulting bed files were performed using bedops. GRCh38 positions were mapped back to GRCh37/hg19 using the online liftOver tool (https://genome.ucsc.edu/cgi-bin/hgLiftOver)

**Differential gene expression in humanized mouse model of hyperuricemia**. To investigate whether the closest genes of the urate-associated CpGs might be differentially expressed in hyperuricemic condition, we obtained gene expression data from a humanized mouse model of hyperuricemia. The orthologous mouse variant to the human ABCG2 Q141K (ABCG2 Q140K) was knocked in to the endogenous Abcg2 locus using CRISPR/Cas9 on a C57BL/6J background, resulting in hyperuricemia in adult male Q140K+/+ mice as described previously[64]. Animal studies were performed in adherence to the NIH Guide for the Care and Use of Laboratory Animals and approved by the University of Maryland School of Medicine Institutional Animal Care and Use Committee. Mice were housed in groups of two to five per cage on a 12:12 h light/dark cycle; with lights on at 6 a.m., only male mice were used for the analysis. Animal organs were harvested and preserved in RNALater (Sigma R0901) and gradually cooled from 4 °C to −80 °C. RNA was isolated using the RNeasy Plus Mini Kit (Qiagen 74134) per the manufacturer's protocol. RNA was suspended in molecular biology grade water and then quantified with a CLARIOstar plate reader and stored at −80 °C. Illumina Sequencing Libraries were prepared using manufacturer's protocol for NEB Ultra II Directional RNA Library Prep kit with poly-A enrichment (NEB E7760). Samples were sequenced on four flowcell lanes of an Illumina HiSeq4000 75 bp paired end run. Five samples were sequenced in each flowcell lane. Samples were grouped per lane by tissue type. Sample quality assessment, RNA-Seq library preparation, and sequencing were performed by the Genomics Resource Center at the University of Maryland School of Medicine. RNA-Seq data was stored and analyzed on BasePair (https://app.basepairtech.com/) for expression count (STAR) and differential expression (DESeq2) analyses. Mouse gene names were mapped to human gene names in HUGO Gene Nomenclature using the Mouse Genome Informatics database. FDR for differential expression was calculated using the Benjamini–Hochberg method. We consider FDR < 0.05 as being significant.

**Enrichment analyses**. To inform the potential functional effects of the urate-associated CpGs, we assessed the enrichment of these CpGs in sites of DNase I or histone modification (H3K4me1, H3K4me3, H3K9me3, H3K27me3), gene sets based on GO terms and pathways in the KEGG and Reactome databases[92–95]. The enrichment analyses of DNase I or histone modification were performed using a local version of eForge version 2.0 (experimentally derived Functional element Overlap analysis of ReGions from EWAS)[96]. Briefly, eForge used date sources from either the ENCODE (125 samples) or Roadmap Epigenomics (299 samples) projects generated by the Hotspot method[97–99]. The overlaps between the input CpGs

and the data sources were compared with those from 1000 random sets of CpGs with matching gene relationship and CpG island annotation. FDR was obtained based on the binomial distribution and the Benjamini–Yekutieli method for multiple testing correction. Our input included 1018 urate-associated CpGs with $p < 1E-05$ in the combined meta-analysis of the discovery and replication cohorts. We performed 10,000 resampling runs with an active proximity filter and considered FDR < 0.01 as significant, i.e., enrichment >99 percentile.

Enrichment in gene sets or pathways was performed using the methylGSA package and R version 3.6.1[100]. The enrichment test method (methylglm) was a functional class scoring method implemented using logistic regression accounting for the number of CpG sites per gene and the autosomal background that overlaps the HM450K and EPIC arrays. Gene sets or pathways with 100–500 genes were tested (default setting). We considered a gene set or pathway to be significantly enriched at FDR < 0.05 correcting for multiple testing within each database using the Benjamini and Hochberg method[101].

**Relationship of urate-associated CpGs with cardiometabolic traits**. Our prior work on the genetics of serum urate revealed significant genetic correlations between serum urate and many cardiometabolic traits and showed that genes mapping into urate-associated loci were highly enriched for expression in kidney and liver[3]. Other than gout, cardiometabolic traits with high genetic correlations ($r^2 > 0.35$) included triglycerides, HDL cholesterol, diabetes, fasting insulin, hypertension, and BMI. To gain insights on whether these relationships were also observed when studying DNA methylation in blood, we investigated whether the urate-associated CpGs were also identified in EWAS of the respective cardiometabolic traits as well as in EWAS of eGFR and liver enzymes. We also included CRP, given that CRP is a marker of inflammation, a component of metabolic syndrome[38]. When more than one EWAS were available for a trait, EWAS were prioritized for the larger sample size. The primary focus was placed on the interpretation of the direction of association of DNA methylation with serum urate and the other traits. The CpGs included in the lookup were those replicated in the EWAS of urate and also considered significant in the EWAS of the other traits based on criteria of the respective study (Supplementary Data 19). All studies included in this lookup controlled for age, sex, and white blood cell type proportions for detecting independent association between DNA methylation and the trait. More details are provided in Supplementary Data 19. When a study included replication and provided meta-analysis results of discovery and replication, the effects from the meta-analysis combining discovery and replication were used.

**Reporting summary**. Further information on research design is available in the Nature Research Reporting Summary linked to this article.

**Disclaimer**. The views expressed in this manuscript are those of the authors and do not necessarily represent the views of the National Heart, Lung, and Blood Institute, the National Institutes of Health, or the US Department of Health and Human Services.

## Data availability
The summary statistics from the meta-analysis are available from the CKDGen Consortium website (https://ckdgen.imbi.uni-freiburg.de). Additional data and programming code that support the findings of this study such as code used to call on software are available from the authors upon request.

## Code availability
The script for generating the phenotypes used in the EWAS is available via GitHub (https://github.com/genepi-freiburg/ckdgen-pheno-ewas)[102]. EWAS QC, meta-analysis, and postprocessing were implemented in R v4.0.1 using metafor v2.4.0, qqman v0.1.4, limma v3.42.2, openxlsx v4.1.5, car v3.0.8, bacon v1.16.0, mutoss v0.1.12, methylGSA v1.6.1, ggplot2 v3.3.3, SeSAMe v1.10.5, rmeta v3.0, mediation v4.5.0, TwoSampleMR R v0.5.6, and Genome-wide Complex Trait Analysis (GCTA) v1.93.2beta.

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

## Acknowledgements

Study-specific acknowledgements and funding sources are listed as Supplementary Note 1.

## Author contributions

Drafting of manuscript: Anna Köttgen, Pascal Schlosser, Alexander Teumer, Chris H. L. Thio, Owen M. Woodward, Adrienne Tin. Bioinformatics: Shreeram Akilesh, Nasir A. Aziz, Xu Gao, Peter Henneman, Anselm Hoppmann, Roby Joehanes, Christine Ladd-Acosta, Hongbo Liu, Ake T. Lu, Pamela R. Matias-Garcia, Daniel L. McCartney, Karlijn A.C. Meeks, Christoph Nowak, Pascal Schlosser, Andrea Venema, Antoine Weihs. Critical review of manuscript: Adebowale A. Adeyemo, Charles Agyemang, Nasir A. Aziz, Andrea Baccarelli, Hermann Brenner, Jan Bressler, Monique M. B. Breteler, Cristian Carmeli, Layal Chaker, Josef Coresh, Adolfo Correa, Simon R. Cox, Graciela E. Delgado, Kai-Uwe Eckardt, Arif B. Ekici, Karlhans Endlich, James S. Floyd, Xu Gao, Allan C. Gelber, Mohsen Ghanbari, Christian Gieger, Philip Greenland, Megan L. Grove, Sarah E. Harris, Peter Henneman, Christian Herder, Lifang Hou, Silva Kasela, Marcus E. Kleber, Wolfgang Koenig, Jaspal S. Kooner, Florian Kronenberg, Anna Köttgen, Christine Ladd-Acosta, Daniel Levy, Lars Lind, Marie Loh, Stefan Lorkowski, Riccardo E. Marioni, Pamela R. Matias-Garcia, Daniel L. McCartney, Karlijn A. C. Meeks, Lili Milani, Winfried März, Matthias Nauck, Christoph Nowak, Holger Prokisch, Bruce M. Psaty, Alex P. Reiner, Pascal Schlosser, Joel Schwartz, Sanaz Sedaghat, Jennifer A. Smith, Nona Sotoodehnia, Hannah R. Stocker, Johan Sundström, Katalin Susztak, Alexander Teumer, Chris H. L. Thio, Adrienne Tin, Uwe Völker, Melanie Waldenberger, Juliane Winkelmann, Yinan Zheng, Johan Ärnlöv. Statistical methods and analysis: Nasir A. Aziz, Cristian Carmeli, Graciela E. Delgado, Eliza Fraszczyk, Xu Gao, Sahar Ghasemi, Franziska Grundner-Culemann, Gibran Hemani, Peter Henneman, Anselm Hoppmann, Steve Horvath, Mikko A. Hurme, Shih-Jen Hwang, Roby Joehanes, Silva Kasela, Marcus E. Kleber, Anna Köttgen, Brigitte Kühnel, Christine Ladd-Acosta, Terho Lehtimäki, Dan Liu, Hongbo Liu, Marie Loh, Ake T. Lu, Pamela R. Matias-Garcia, Daniel L. McCartney, Karlijn A. C. Meeks, Josine L. Min, Pashupati P. Mishra, Christoph Nowak, Scott M. Ratliff, Pascal Schlosser, Sanaz Sedaghat, Brenton R. Swenson, Alexander Teumer, Chris H. L. Thio, Adrienne Tin, Andrea Venema, Antoine Weihs, Zhi Yu, Wei Zhao, Yinan Zheng. Functional studies: Victoria L. Halperin Kuhns, Hongbo Liu, Katalin Susztak, Owen M. Woodward. Phenotyping: Charles Agyemang, Murielle Bochud, Hermann Brenner, Monique M. B. Breteler, Simon R. Cox, James S. Floyd, Sarah E. Harris, Christian Herder, Sharon L. R. Kardia, Silva Kasela, Wolfgang Koenig, Anna Köttgen, Joyce B. J. van Meurs, Annette Peters, Bruce M. Psaty, Scott M. Ratliff, Johan Ärnlöv. Interpretation of results: Shreeram Akilesh, Nasir A. Aziz, Andrea Baccarelli, Hermann Brenner, Jan Bressler, Layal Chaker, James S. Floyd, Xu Gao, Allan C. Gelber, Anna Köttgen, Christine Ladd-Acosta, Daniel Levy, Pamela R. Matias-Garcia, Christoph Nowak, Bruce M. Psaty, Pascal Schlosser, Joel Schwartz, Ben Schöttker, Nona Sotoodehnia, Katalin Susztak, Alexander Teumer, Chris H. L. Thio, Adrienne Tin, Owen M. Woodward. Study design of an individual contributing study: Adebowale A. Adeyemo, Charles Agyemang, Andrea Baccarelli, Murielle Bochud, Hermann Brenner, Monique M. B. Breteler, Layal Chaker, Adolfo Correa, Simon R. Cox, Mohsen Ghanbari, Christian Gieger, Gibran Hemani, Peter Henneman, Lifang Hou, Sharon L. R. Kardia, Jaspal S. Kooner, Daniel Levy, Ake T. Lu, Riccardo E. Marioni, Karlijn A.C. Meeks, Josine L. Min, Winfried März, Annette Peters, Holger Prokisch, Bruce M. Psaty, Alex P. Reiner, Joel Schwartz, Jennifer A. Smith, Harold Snieder, Jana V. van Vliet-Ostaptchouk, Juliane Winkelmann, Bruce H.R. Wolffenbuttel, Johan Ärnlöv. Subject recruitment: Charles Agyemang, Nasir A. Aziz, Andrea Baccarelli, Murielle Bochud, Hermann Brenner, Simon R. Cox, Jaspal S. Kooner, Lars Lind, Karlijn A. C. Meeks, Lili Milani, Matthias Nauck, Annette Peters, Joel Schwartz. Methylation assessment: Charles Agyemang, Andrea Baccarelli, Hermann Brenner, James S. Floyd, Xu Gao, Megan L. Grove, Xin Gào, Sarah E. Harris, Peter Henneman, Steve Horvath, Lifang Hou, Mikko A. Hurme, Silva Kasela, Marcus E. Kleber, Stefan Lorkowski, Daniel L. McCartney, Joyce B. J. van Meurs, Lili Milani, Jennifer A. Smith, Nona Sotoodehnia, Silvia Stringhini, Alexander Teumer, Adrienne Tin, Jana V. van Vliet-Ostaptchouk, Melanie Waldenberger, Juliane Winkelmann, Bruce H. R. Wolffenbuttel, Wei Zhao, Yinan Zheng. Management of an individual contributing study: Adebowale A. Adeyemo, Nasir A. Aziz, Andrea Baccarelli, Murielle Bochud, Hermann Brenner, Monique M. B. Breteler, Layal Chaker, Josef Coresh, Tanguy Corre, Adolfo Correa, Simon R. Cox, Christian Gieger, Megan L. Grove, Xin Gào, Sarah E. Harris, Gibran Hemani, Peter Henneman, Lifang Hou, Mikko A. Hurme, Sharon L. R. Kardia, Marcus E. Kleber, Wolfgang Koenig, Jaspal S. Kooner, Anna Köttgen, Terho Lehtimäki, Daniel Levy, Lars Lind, Donald M. Lloyd-Jones, Marie Loh, Riccardo E. Marioni, Karlijn A. C. Meeks, Joyce B. J. van Meurs, Lili Milani, Josine L. Min, Pashupati P. Mishra, Winfried März, Matthias Nauck, Annette Peters, Bruce M. Psaty, Olli T. Raitakari, Alex P. Reiner, Joel Schwartz, Ben Schöttker, Jennifer A. Smith, Harold Snieder, Hannah R. Stocker, Johan Sundström, Jana V. van Vliet-Ostaptchouk, Uwe Völker, Melanie Waldenberger, Juliane Winkelmann, Bruce H. R. Wolffenbuttel.

## Funding

## Competing interests

J. Ärnlöv has served on advisory boards for AstraZeneca and Boehringer Ingelheim, and has received lecturing fees from AstraZeneca and Novartis, all unrelated to the present project. J. Coresh has grants from NIH and is a consultant to healthy.io. J. S. Floyd has consulted for Shionogi Inc. C. Herder reports personal fees from Sanofi and Lilly and grant support from Sanofi outside the submitted work. M. E. Kleber is employed with SYNLAB Holding Deutschland GmbH. O. Woodward has grants from AstraZeneca outside the submitted work. W. Koenig reports personal fees from AstraZeneca, Novartis, Pfizer, The Medicines Company, DalCor, Kowa, Amgen, Corvidia, Daichii-Sankyo, Genentech, Novo Nordisk, Omeicos, Esperion, Berlin-Chemie, Sanofi, and Bristol-Myers Squibb and grants and non-financial support from Abbott, Roche Diagnostics, Beckmann, and Singulex outside the submitted work. S. Lorkowski reports grants and personal fees from Akcea Therapeutics Germany, and personal fees from amedes, AMGEN, Berlin-Chemie, Boehringer Ingelheim Pharma, Daiichi Sankyo, Lilly Deutschland, MSD Sharp & Dohme, Novo Nordisk Pharma, Roche Pharma, Sanofi-Aventis, Synlab Holding Deutschland, Unilever, and Upfield, all outside the submitted work. R. E. Marioni has received payment from Illumina for presentations. W. März reports grants from Siemens Healthineers, grants and personal fees from Aegerion Pharmaceuticals, grants and personal fees from AMGEN, grants from Astrazeneca, grants and personal fees from Sanofi, grants and personal fees from Alexion Pharmaceuticals, grants and personal fees from BASF, grants and personal fees from Abbott Diagnostics, grants and personal fees from Numares AG, grants and personal fees from Berlin-Chemie, grants and personal fees from Akzea Therapeutics, grants from Bayer Vital GmbH, grants from bestbion dx GmbH, grants from Boehringer Ingelheim Pharma GmbH Co KG, grants from Immundiagnostik GmbH, grants from Merck Chemicals GmbH, grants from MSD Sharp and Dohme GmbH, grants from Novartis Pharma GmbH, grants from Olink Proteomics, other from Synlab Holding Deutschland GmbH, all outside the submitted work. B. Psaty serves on the Steering Committee of the Yale Open Data Access Project funded by Johnson & Johnson. J. Sundström reports ownership in companies providing services to Itrim, Amgen, Janssen, Novo Nordisk, Eli Lilly, Boehringer, Bayer, Pfizer, and AstraZeneca, outside the submitted work. B. Kühnel is currently an employee of Regeneron Genetics Center. All other authors have nothing to declare.

## Additional information

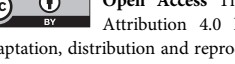

Adrienne Tin [1,2,98 ✉], Pascal Schlosser [2,3,98], Pamela R. Matias-Garcia [4,5,6], Chris H. L. Thio [7], Roby Joehanes [8,9], Hongbo Liu [10], Zhi Yu [11,12,13], Antoine Weihs [14], Anselm Hoppmann [3], Franziska Grundner-Culemann [3], Josine L. Min [15,16], Victoria L. Halperin Kuhns [17], Adebowale A. Adeyemo [18], Charles Agyemang [19], Johan Ärnlöv [20,21], Nasir A. Aziz [22,23], Andrea Baccarelli [24], Murielle Bochud [25], Hermann Brenner [26,27,28,29], Jan Bressler [30], Monique M. B. Breteler [22,31], Cristian Carmeli [25,32], Layal Chaker [33,34], Josef Coresh [2], Tanguy Corre [25], Adolfo Correa [1], Simon R. Cox [35], Graciela E. Delgado [36], Kai-Uwe Eckardt [37,38], Arif B. Ekici [39], Karlhans Endlich [40,41], James S. Floyd [42,43,44], Eliza Fraszczyk [7], Xu Gao [24,45], Xīn Gào [26], Allan C. Gelber [2,46], Mohsen Ghanbari [33], Sahar Ghasemi [14,41,47], Christian Gieger [4,5], Philip Greenland [48], Megan L. Grove [30], Sarah E. Harris [35], Gibran Hemani [15,16], Peter Henneman [49], Christian Herder [50,51,52], Steve Horvath [53,54], Lifang Hou [48], Mikko A. Hurme [55], Shih-Jen Hwang [8,56], Sharon L. R. Kardia [57], Silva Kasela [58], Marcus E. Kleber [36,59], Wolfgang Koenig [60,61,62], Jaspal S. Kooner [63,64,65], Florian Kronenberg [66], Brigitte Kühnel [4,5], Christine Ladd-Acosta [2], Terho Lehtimäki [67,68,69], Lars Lind [70], Dan Liu [22], Donald M. Lloyd-Jones [48], Stefan Lorkowski [71,72], Ake T. Lu [53], Riccardo E. Marioni [73], Winfried März [36,72,74,75], Daniel L. McCartney [73], Karlijn A. C. Meeks [18,19], Lili Milani [58], Pashupati P. Mishra [67,68,69], Matthias Nauck [41,76], Christoph Nowak [20], Annette Peters [5,77], Holger Prokisch [78,79], Bruce M. Psaty [42,43,44,80], Olli T. Raitakari [81,82,83], Scott M. Ratliff [57], Alex P. Reiner [43], Ben Schöttker [26,27], Joel Schwartz [84], Sanaz Sedaghat [85], Jennifer A. Smith [57,86], Nona Sotoodehnia [44], Hannah R. Stocker [26,27], Silvia Stringhini [25], Johan Sundström [70,87], Brenton R. Swenson [44,88], Joyce B. J. van Meurs [34], Jana V. van Vliet-Ostaptchouk [89], Andrea Venema [49], Uwe Völker [41,90], Juliane Winkelmann [78,91,92,93], Bruce H. R. Wolffenbuttel [89], Wei Zhao [57], Yinan Zheng [48], The Estonian Biobank Research Team*, The Genetics of DNA Methylation Consortium*, Marie Loh [94,95], Harold Snieder [7], Melanie Waldenberger [4,5,61], Daniel Levy [8,9], Shreeram Akilesh [96], Owen M. Woodward [17], Katalin Susztak [10], Alexander Teumer [41,47,97,99] & Anna Köttgen [2,3,99 ✉]

[1]Department of Medicine, University of Mississippi Medical Center, Jackson 39216 MS, USA. [2]Department of Epidemiology, Johns Hopkins Bloomberg School of Public Health, Baltimore, MD, USA. [3]Institute of Genetic Epidemiology, Faculty of Medicine and Medical Center, University of Freiburg, Freiburg, Germany. [4]Research Unit Molecular Epidemiology, Helmholtz Zentrum München, German Research Center for Environmental Health, Neuherberg D-85764 Bavaria, Germany. [5]Institute of Epidemiology, Helmholtz Zentrum München, German Research Center for Environmental Health, Neuherberg D-85764 Bavaria, Germany. [6]TUM School of Medicine, Technical University of Munich, Munich, Germany. [7]Department of Epidemiology, University of Groningen, University Medical Center Groningen, Groningen, the Netherlands. [8]Framingham Heart Study, Framingham, MA, USA. [9]Population Sciences Branch, National Heart, Lung, and Blood Institute, National Institutes of Health, Bethesda, MD, USA. [10]Department of Medicine and Genetics, University of Pennsylvania Perelman School of Medicine, Philadelphia 19104 PA, USA. [11]Program in Medical and Population Genetics, Broad Institute, Cambridge, MA, USA. [12]Cardiovascular Research Center, Massachusetts General Hospital, Boston, MA, USA. [13]Center for Genomic Medicine, Massachusetts General Hospital, Boston, MA, USA. [14]Department of Psychiatry and Psychotherapy, University Medicine Greifswald, Greifswald, Germany. [15]MRC Integrative Epidemiology Unit, University of Bristol, Bristol, UK. [16]Population Health Sciences, Bristol Medical School, University of Bristol, Bristol, UK. [17]Department of Physiology, University of Maryland School of Medicine, Baltimore, MD, USA. [18]Center for Research on Genomics and Global Health, National Human Genome Research Institute, National Institutes of Health, Bethesda, MD, USA. [19]Department of Public and Occupational Health, Amsterdam Public Health Research Institute, Amsterdam University Medical Centers, University of Amsterdam, 1105 AZ Amsterdam, the Netherlands. [20]Department of Neurobiology, Care Sciences and Society (NVS), Family Medicine and Primary Care Unit, Karolinska Institutet, Huddinge, Sweden. [21]School of Health and Social Studies, Dalarna University, Falun, Sweden. [22]Population Health Sciences, German Centre for Neurodegenerative Diseases (DZNE), Bonn, Germany. [23]Department of Neurology, Faculty of Medicine, University of Bonn, Bonn, Germany. [24]Laboratory of Environmental Precision Health, Mailman School of Public Health, Columbia University, New York, NY, USA. [25]Center for Primary Care and Public Health (Unisanté), University of Lausanne, Lausanne, Switzerland. [26]German Cancer Research Center (DKFZ), Division of Clinical Epidemiology and Aging Research, Heidelberg, Germany. [27]Network Aging Research, Heidelberg University, Heidelberg, Germany. [28]Division of Preventive Oncology, German Cancer Research Center (DKFZ) and National Center for Tumor Diseases (NCT), Heidelberg, Germany. [29]German Cancer Consortium, German Cancer Research Center (DKFZ), Heidelberg, Germany. [30]Human Genetics Center, Department of Epidemiology, Human Genetics and Environmental Sciences, School of Public Health, The University of Texas Health Science Center at Houston, Houston 77030 TX, USA. [31]Institute for Medical Biometry, Informatics and Epidemiology (IMBIE), Faculty of Medicine, University of Bonn, Bonn, Germany. [32]Population Health Laboratory, University of Fribourg, Fribourg, Switzerland. [33]Department of Epidemiology, Erasmus University Medical Center, Rotterdam, the Netherlands. [34]Department of Internal Medicine, Erasmus Medical Center, Rotterdam, the Netherlands. [35]Lothian Birth Cohorts Group, Department of Psychology, The University of Edinburgh, 7 George Square, Edinburgh EH8 9JZ, UK. [36]Vth Department of Medicine, Medical Faculty Mannheim, Heidelberg University, Mannheim, Germany. [37]Department of Nephrology and Hypertension, University of Erlangen-Nürnberg, Erlangen, Germany. [38]Department of Nephrology and Medical Intensive Care, Charité – Universitätsmedizin Berlin, Berlin, Germany. [39]Institute of Human Genetics, Friedrich-Alexander-UniversitätErlangen-Nürnberg, 91054 Erlangen, Germany. [40]Department of Anatomy and Cell Biology, University Medicine Greifswald,

Greifswald, Germany. [41]DZHK (German Centre for Cardiovascular Research), Partner Site Greifswald, Greifswald, Germany. [42]Department of Medicine, University of Washington, Seattle 98101 WA, USA. [43]Department of Epidemiology, University of Washington, Seattle 98101 WA, USA. [44]Cardiovascular Health Research Unit, University of Washington, Seattle 98101 WA, USA. [45]Department of Occupational and Environmental Health Sciences, School of Public Health, Peking University, Beijing, China. [46]Department of Medicine, Johns Hopkins School of Medicine, Baltimore, MD, USA. [47]Institute for Community Medicine, University Medicine Greifswald, Greifswald, Germany. [48]Department of Preventive Medicine, Northwestern University Feinberg School of Medicine, Chicago, IL, USA. [49]Department of Clinical Genetics, Amsterdam Reproduction & Development Research Institute, Amsterdam University Medical Centres, University of Amsterdam, Amsterdam, the Netherlands. [50]Institute for Clinical Diabetology, German Diabetes Center, Leibniz Center for Diabetes Research at Heinrich Heine University Düsseldorf, Düsseldorf, Germany. [51]German Center for Diabetes Research, Munich-Neuherberg, Germany. [52]Division of Endocrinology and Diabetology, Medical Faculty, Heinrich Heine University Düsseldorf, Düsseldorf, Germany. [53]Department of Human Genetics, David Geffen School of Medicine, University of California Los Angeles, Los Angeles 90095 CA, USA. [54]Biostatistics, Fielding School of Public Health, UCLA, Los Angeles, CA, USA. [55]Department of Microbiology and Immunology, Faculty of Medicine and Health Technology, Tampere University, Tampere 33014, Finland. [56]Division of Intramural Research, Population Sciences Branch, National Heart, Lung, and Blood Institute, National Institutes of Health, Bethesda, MD, USA. [57]Department of Epidemiology, School of Public Health, University of Michigan, Ann Arbor 48109 MI, USA. [58]Estonian Genome Centre, Institute of Genomics, University of Tartu, Tartu, Estonia. [59]SYNLAB MVZ Humangenetik Mannheim, Mannheim, Germany. [60]Deutsches Herzzentrum München, Technische Universität München, Munich, Germany. [61]DZHK (German Centre for Cardiovascular Research), Partner site Munich Heart Alliance, Munich, Germany. [62]Institute of Epidemiology and Medical Biometry, University of Ulm, Ulm, Germany. [63]National Heart and Lung Institute, Imperial College London, London, UK. [64]Department of Cardiology, Ealing Hospital, London North West Healthcare NHS Trust, Southall, UK. [65]Imperial College Healthcare NHS Trust, London, UK. [66]Institute of Genetic Epidemiology, Medical University of Innsbruck, Innsbruck, Austria. [67]Department of Clinical Chemistry, Faculty of Medicine and Health Technology, Tampere University, Tampere, Finland. [68]Finnish Cardiovascular Research Centre, Faculty of Medicine and Health Technology, Tampere University, Tampere, Finland. [69]Department of Clinical Chemistry, Fimlab Laboratories, Tampere, Finland. [70]Department of Medical Sciences, Uppsala University, Uppsala, Sweden. [71]Institute of Nutritional Sciences, Friedrich Schiller University Jena, Jena, Germany. [72]Competence Cluster for Nutrition and Cardiovascular Health (nutriCARD) Halle-Jena-Leipzig, Jena, Germany. [73]Centre for Genomic and Experimental Medicine, Institute of Genetics and Cancer, University of Edinburgh, Edinburgh EH4 2XU, UK. [74]Synlab Academy, SYNLAB Holding Deutschland GmbH, Mannheim and Augsburg, Germany. [75]Clinical Institute of Medical and Chemical Laboratory Diagnostics, Medical University of Graz, Graz, Austria. [76]Institute of Clinical Chemistry and Laboratory Medicine, University Medicine Greifswald, Greifswald, Germany. [77]Ludwig-Maximilians Universität München, Munich, Germany. [78]Institute of Human Genetics, Klinikum rechts der Isar, Technische Universität München, Munich, Germany. [79]Department of Computational Health, Institute of Neurogenomics, Helmholtz Zentrum München, Munich, Germany. [80]Department of Health Services, University of Washington, Seattle 98101 WA, USA. [81]Research Centre of Applied and Preventive Cardiovascular Medicine, University of Turku, Turku, Finland. [82]Department of Clinical Physiology and Nuclear Medicine, Turku University Hospital, Turku, Finland. [83]Centre for Population Health Research, University of Turku and Turku University Hospital, Turku, Finland. [84]Department of Environmental Health, Harvard T.H. Chan School of Public Health, Boston, MA, USA. [85]Division of Epidemiology and Community Health, School of Public Health, University of Minnesota, Minneapolis, MN, USA. [86]Survey Research Center, Institute for Social Research, University of Michigan, Ann Arbor, MI, USA. [87]The George Institute for Global Health, University of New South Wales, Sydney, NSW, Australia. [88]Institute for Public Health Genetics, University of Washington, Seattle, WA, USA. [89]Department of Endocrinology, University of Groningen, University Medical Center Groningen, Groningen, the Netherlands. [90]Interfaculty Institute for Genetics and Functional Genomics, University Medicine Greifswald, Greifswald, Germany. [91]Institute of Neurogenomics, Helmholtz Zentrum München, Munich, Germany. [92]Chair Neurogenetics, Klinikum rechts der Isar, Technische Universität München, Munich, Germany. [93]Munich Cluster for Systems Neurology (SyNergy), Munich, Germany. [94]Lee Kong Chian School of Medicine, Nanyang Technological University, Singapore, Singapore. [95]Department of Epidemiology and Biostatistics, Imperial College London, London, UK. [96]Department of Laboratory Medicine and Pathology, University of Washington, Seattle, WA, USA. [97]Department of Population Medicine and Lifestyle Diseases Prevention, Medical University of Bialystok, Bialystok, Poland. [98]These authors contributed equally: Adrienne Tin, Pascal Schlosser. [99]These authors jointly supervised this work: Alexander Teumer, Anna Köttgen. *Lists of authors and their affiliations appear at the end of the paper. ✉email: atin@umc.edu; anna.koettgen@uniklinik-freiburg.de

## The Estonian Biobank Research Team

Lili Milani [58]

A full list of members and their affiliations appears in the Supplementary Information.

## The Genetics of DNA Methylation Consortium

Josine L. Min [15,16] & Gibran Hemani [15,16]

