## [Peer Review File · Nature Communications]

Epigenome-wide association study of serum urate reveals insights into urate co-regulation and the SLC2A9 locusReviewers' Comments:

Reviewer #1:

Remarks to the Author:

The authors conducted the largest EWAS for serum urate levels against CpG, discovering 100. The major locus was SLC2A9, where a complicated pattern of association was detected.

I have two major points and a series of minor points.

Major comments

1. The authors conducted Mendelian randomization to attempt to disentangle the relationship between SU and altered CpG methylation at SLC2A9. I found it striking that, for each CpG, there were >20 apparently independent cis-meQTLs for the MR testing for a causal role of CpG methylation on urate. Is this typical? (Min et al reported a median of 2.) Is there any chromosome connectivity data to give more confidence to the cis-meQTLs. The authors used a LD-clumping threshold of $r^2 < 0.2$. I would like to see a lower LD-clumping threshold used (eg < 0.05) i.e. are these meQTLs truly genetically independent, this could be checked by conditional analysis. Looking through the list for cg13841979 (Fig S5A) ~6 were multi-allelic. How does the harmonization function of 2sampleMR cope with this? What MAF cut-off was used? (I looked at Min et al, which has not been peer-reviewed, but this wasn't helpful, I could not find a substantive methods section.) Finally, the contributing studies to Min et al only reported back on meQTLs with $p, 10^{-5}$. Could this create bias / instability, and be a cause of the very low Phet values in the MR (Table 1). While I agree with the statement lines 328-9 re complex situation at SLC2A9, these low Phet values require more interrogation. In summary, this section of work did not 'feel right' to me. The authors should redo with more careful curation of IV SNPs.

With respect to the SU vs CpG methylation MR only the FHS was used. (Why was ARIC not used as well.) Excluding the SLC2A9 variant what was the power? (I imagine low.) (The SLC2A9 variant should really be excluded as it is from the same locus of the major CpG effects.) How was family structure accounted for? Unless an adequately powered MR analysis can be done sans SLC2A9 then this analysis, perhaps, should not be presented. Or, at least, more cautiously interpreted.

2. I was surprised that the authors did not weave 'training' of the innate immune system to be more responsive to MSU crystals by soluble urate into the interpretation of their results. This important phenomenon was not mentioned at all. Admittedly soluble urate has not yet been shown to change methylation status, although it has with other training exposures. A schema can be developed whereby increased urate -> transported into monocytes by SLC2A9 -> altered methylation at SLC2A9 and other genes -> may increase urate levels via increased renal reuptake and increase responsiveness to MSU crystals.

As one example the perplexing heritability data on pp 11-12 could be interpreted via a training effect of soluble urate. (This possibility is consistent with the sentence lines 293-5.) Training could lead to a direct correlation between SU and methylation. There are other places in the manuscript, including discussion where training as a mechanism could be woven in.

I note that the urate-associated CpG genes encode transporters that could conceivably provide substrate for epigenomic modification.....

Other comments.

1. In the context of the correlation with other metabolic traits it is important to point out that there is no robust evidence by MR for soluble urate being causal of these traits.

2. Line 253 'median of the mean age'.

3. In the mediation analysis there were only 3 independent variants reported at SLC2A9 in Tin et al (Table S5). Also, are these variants truly independent, as Tin et al 2019 used a genomic distance cut-off, there was no LD-clumping or conditional analysis. (And the SNPs in the current study were different than those reported in Tin et al 2019 - are they surrogates?)

4. Something odd re references line 528.

5. Line 557, the phrase 'complexity of associations of SNPs in the SLC2A9 locus' did not make sense to me.
6. Para beginning line 562. Of course the most associated genes are going to support the pathway analysis as they were used to generate the pathway analysis. This is a circular argument.
7. Figures 2 and 3 - what is the LD measure used?

Reviewer #2:

Remarks to the Author:

In this interesting transethnic meta-analysis, Tin et al., have utilized 24 cohort studies to examine epigenome wide associations of serum urate. The authors have performed rigorous statistical analyses and used multiple layers of complementary evidence to support the validity of the findings. The authors have discussed on the complexity of these associations, biological plausibility of the results, and the associations of urate-associated methylation markers with other complex traits. The paper is well-written, clear description of methods and with noteworthy findings. I have a few minor comments.

Line 254: It will be better to mention that the "Normative aging study (NAS)" only included men in the replication cohort. Other cohorts included both men and women.

Line 261: The authors have identified 140 significant CpGs (discovery dataset) associated with serum urate levels in the trans-ethnic meta-analysis including European, African American, South Asian and Sub-Saharan African Ancestries and 99 significant CpGs in the replication dataset. Did the authors evaluate whether the urate-associated methylation markers were significant across the ancestries included in the study, and whether there was any ancestry-specific DNA methylation marker? Additional information on the ancestry specific markers would be good to see in the supplementary information.

Line 322: The causal effect estimates on serum urate (Table 1) results ranged from 0.16 to 0.30 mg/dl per standard deviation... instead of "0.15". Minor typo to correct.

Line 324-325: Similar to the above sentence (Table 1), "effect sizes ranging from -0.38 to -0.51 mg/dl per SD.. instead of "-0.37". Minor typo to correct.

Figure 2. Please include A and B in the main figures for PHGDH and SLC1A5. You have described separately as A and B in the legends but is missing in the figures.

Figure 3, line 1026: Legend typo error for the "HM450K" annotation file.

Reviewer #3:

Remarks to the Author:

This is a remarkably dense paper, which I have had to read several times to understand what is going on. The authors are to be commended that all the analyses are stringently carried out and unbiased. However, I feel that more effort is necessary to communicate the biological messages from the results.

Methylation is a dynamic process that often reflects the regulatory state of particular loci. It is very rarely heritable, for example with imprinted loci. The authors are aware that apparent heritability may be attributed to underlying SNPS affecting transcription binding sites. Their investigation of SNP/DMR interactions is worthy but the train of the analysis is difficult to follow.

The biological meat of the study is in table 4 of the supplementary information where CpG/DMRs and their associated genes are listed. There are some really interesting genes here that are unmentioned in the text. I would put an abbreviated form of this into the main paper and include columns with a brief description of putative gene function and an indicator of the presence of SNPs associated with the trait.

I would provide detailed maps, such as figure 2, of the most important of these loci, including recognised genomic features such as CpG islands and known regulatory elements, and the position of trait-associated SNPs.

I find the description of Mendelian Randomisation choked with jargon and really really difficult to understand. Some simple explanation would make the paper far more accesible to non-statistical readers.

A major problem is that the DNA for DMR identification came from peripheral blood, which is a mixture of different cell types with quite different functions and many cell-specific DMRs. This means that DMRs associated with any of the metabolic syndrome traits may simply refelect changes in the elements of the white cell count. Some effort needs to be made to deconvolute this. At the simplest level thsi could be detecting correlations with WCC subsets (neutrophils, eosinophils etc. etc.), but the use of invariant CpG markers of cell lineages or WGCNA analyses are also possible

Reviewer #1 (Remarks to the Author):

The authors conducted the largest EWAS for serum urate levels against CpG, discovering 100. The major locus was SLC2A9, where a complicated pattern of association was detected.

I have two major points and a series of minor points.

Major comments

1. The authors conducted Mendelian randomization to attempt to disentangle the relationship between SU and altered CpG methylation at SLC2A9. I found it striking that, for each CpG, there were >20 apparently independent cis-meQTLs for the MR testing for a causal role of CpG methylation on urate. Is this typical? (Min et al reported a median of 2.) Is there any chromosome connectivity data to give more confidence to the cis-meQTLs. The authors used a LD-clumping threshold of $r^2 < 0.2$. I would like to see a lower LD-clumping threshold used (eg < 0.05) i.e. are these meQTLs truly genetically independent, this could be checked by conditional analysis. Looking through the list for cg13841979 (Fig S5A) ~6 were multi-allelic. How does the harmonization function of 2sampleMR cope with this? What MAF cut-off was used? (I looked at Min et al, which has not been peer-reviewed, but this wasn't helpful, I could not find a substantive methods section.) Finally, the contributing studies to Min et al only reported back on meQTLs with $p, 10^{-5}$. Could this create bias / instability, and be a cause of the very low Phet values in the MR (Table 1). While I agree with the statement lines 328-9 re complex situation at SLC2A9, these low Phet values require more interrogation. In summary, this section of work did not 'feel right' to me. The authors should redo with more careful curation of IV SNPs.

With respect to the SU vs CpG methylation MR only the FHS was used. (Why was ARIC not used as well.) Excluding the SLC2A9 variant what was the power? (I imagine low.) (The SLC2A9 variant should really be excluded as it is from the same locus of the major CpG effects.) How was family structure accounted for? Unless an adequately powered MR analysis can be done sans SLC2A9 then this analysis, perhaps, should not be presented. Or, at least, more cautiously interpreted.

Response: Thank you for this detailed comment, which we have addressed in several ways. First, regarding the r^2 threshold and checking the independence of meQTLs using conditional analysis, we followed the Reviewer's suggestions and use a two-step process to select independent meQTLs. First, we applied an $r^2 < 0.05$ threshold to screen for meQTLs in addition to the other criteria in our original analysis (MAF >1%, association with DNA methylation at $p < 5E-8$, no significant association with potential confounders, and no potential reverse causation). As recommended by the Reviewer, we then conducted conditional analysis for the meQTLs that passed the screening step to obtain the p-value of each meQTL conditioning on all other meQTLs for a given CpG, and only retained those meQTLs with conditional p-value $< 5E-8$ as instruments. Using the

revised criteria, 27 CpGs had 4 or more meQTLs available for MR analysis. In the new results, 4 CpGs in *SLC2A9* had significant causal effects on urate ($p < 1.9E-3 = 0.05/27$), and 1 in *SLC2A9* had significant causal effects on gout. The updated MR results are reported in **Table 2**. Moreover, we have also added two new sets of plots: a) LD plots showing the r^2 between the meQTLs for each CpG; and b) forest plots of the causal effect estimate for each meQTL to help visualize any heterogeneity of the MR results. The forest plots show that the majority of the meQTLs support the significant causal effects. Revisions in the main text: Results: page 11, paragraph 2 to page 13, paragraph 1; Methods: page 30, paragraph 2 to page 33, paragraph 1; **Table 2**, **Figures 4** and **5**, **Supplementary Tables 10** and **11**; **Supplementary Figures 3 to 10**; Supplementary text: page 10, paragraph 1. All page and paragraph numbers refer to the clean version of the manuscript.

Second, the CpGs at *SLC2A9* indeed had more independent meQTLs than other CpGs. The median numbers of meQTLs that met the revised screening criteria were 2 (25th, 75th percentile: 0, 5) among the 92 replicated CpGs outside of *SLC2A9*, and were 20 (25th, 75th percentile: 8, 21) among the 7 CpGs at *SLC2A9*. *SLC2A9* is the GWAS locus with the largest effect size for urate, and multiple studies reported that DNA methylation at some of the CpGs had high heritability.^{1, 2, 3} For example, the 4 CpGs with significant causal effects on urate in our updated analysis had average heritability estimates from 0.39 to 0.93 from 3 studies (**Supplementary Table 9**), meaning that a substantial proportion of the variance in DNA methylation at these sites is due to additive genetic effects. Thus, heritability in this context does not refer to the inheritance across generations.

Third, about multi-allelic SNPs, some meQTLs have additional rare alleles in dbSNP, we only included meQTLs with MAF >1% and having 2 alleles with imputed frequencies that sum to 100% in the GoDMC analysis. The summary statistics used in the MR analyses were based on meta-analyzed GWAS of imputed genotypes after filtering out SNPs with low imputation quality (info score <0.8 for GoDMC and imputation quality <0.6 for the CKDGen Consortium). While some imputed genotypes therefore contain some degree of imprecision, potentially reducing the power to discovery additional meQTLs, this is unlikely to affect our results overall.

Fourth, about the meQTLs available from the GoDMC data, Min *et al.* only included meQTLs with $p < 10^{-5}$ from at least one cohort. This was to screen for meQTLs that would more likely be genome-wide significant in their meta-analysis across all cohorts. It is possible that some meQTLs with p -value $> 10^{-5}$ within each cohort might have achieved genome-wide significance in a meta-analysis, in which case we may have missed some additional meQTLs, as explained in the GoDMC paper that has just been published in *Nature Genetics*.⁴ However, the screening process in GoDMC was independent of urate, and therefore unlikely to result in bias in our MR results.

Fifth, we used the FHS meQTL results for reverse MR instead of those from ARIC because most meQTL data from ARIC were from African-American participants (**Supplementary Table 1**), whereas the GWAS meta-analysis of serum urate used in MR was based on European ancestry (EA) individuals. The analysis in FHS accounted for family structure using linear mixed model as reported in Huan *et al.* Nat Comm 2019.⁵ The FHS data have the additional advantage of being an independent sample from the urate EWAS analysis.

Sixth, regarding power, an F-statistic >10 is an accepted threshold for avoiding weak instrument bias, and our p-value threshold at genome-wide significance (p-value <5E-8), is equivalent to an F-statistic of 30.⁶ Using a tool cited by the Guidelines for Mendelian randomization studies (<https://sb452.shinyapps.io/power/>),⁷ we calculated the minimum detectable effect size for the CpGs that remained significant in the updated analysis assuming 90% power, a significance threshold of 1.9E-3, and the sample sizes that used in our MR analysis. The results are shown in **Reviewer Table 1**. These effect sizes were well below those detected for urate or gout (**Table 2**). Thus, the power calculations support that our MR analyses were well powered.

Reviewer Table 1. Minimum detectable effect size in forward MR analysis of DNA methylation on serum urate or gout.

Outcome	CpG	All meQTLs in primary analysis		Excluding meQTLs with $r^2 > 0.05$ with 5 urate SNPs in EA*	
		# of meQTLs	Min detectable effect size**	# of meQTLs	Min detectable effect size**
Urate	cg02387843	6	0.044	Not enough meQTL for MR analysis (< 4)	
Urate	cg13841979	10	0.039	7	0.104
Urate	cg03725404	8	0.047	Not enough meQTL for MR analysis (< 4)	
Urate	cg11266682	11	0.020	Not enough meQTL for MR analysis (< 4)	
Gout	cg03725404	8	0.918	Not enough meQTL for MR analysis (< 4)	

*The 5 urate SNPs in *SLC2A9* in EA reported in Tin *et al.* 2019: SNP with lowest p-value in EA: rs4447862, independent SNPs identified by GCTA stepwise selection: rs6825187, rs62286563, rs10017305, rs73224492.

** The effect size was estimated for mg/dL per SD of DNA methylation beta value for urate and OR per SD of DNA methylation beta value for gout.

Lastly, regarding urate-associated variants at *SLC2A9* reported from GWAS and the causal relationship between DNA methylation and serum urate levels. After we excluded meQTLs with $r^2 > 0.05$ with the 5 urate SNPs in EA listed in the footnote of **Reviewer Table 1**, cg13841979 had ≥ 4 meQTLs for MR analysis and was nominally significant (p=2.25E-2 based on our primary method, IVW-MRE as

opposed to $p=1.38E-04$ before; **Supplementary Tables 10**). However, we also identified significant mediating effects of cg13841979 for GWAS index SNPs on serum urate (**Supplementary Table 13**). Excluding meQTLs in LD with GWAS index SNPs in the MR analysis amounts to the removal of the mediating effect of the CpG and thus attenuated the causal effect estimate. In summary, both our MR and mediation analyses support the potentially causal and genetic-variant mediating effects of DNA methylation at *SLC2A9* on urate levels. We have clarified these points in the manuscript (Results: page 11, paragraph 2).

2. I was surprised that the authors did not weave 'training' of the innate immune system to be more responsive to MSU crystals by soluble urate into the interpretation of their results. This important phenomenon was not mentioned at all. Admittedly soluble urate has not yet been shown to change methylation status, although it has with other training exposures. A schema can be developed whereby increased urate -> transported into monocytes by *SLC2A9* -> altered methylation at *SLC2A9* and other genes -> may increase urate levels via increased renal reuptake and increase responsiveness to MSU crystals.

As one example the perplexing heritability data on pp 11-12 could be interpreted via a training effect of soluble urate. (This possibility is consistent with the sentence lines 293-5.) Training could lead to a direct correlation between SU and methylation. There are other places in the manuscript, including discussion where training as a mechanism could be woven in.

I note that the urate-associated CpG genes encode transporters that could conceivably provide substrate for epigenomic modification.....

Response: Thank you for raising this point. We have added the following text in the Discussion section: “Prior exposure to serum urate in its soluble or crystal form has been shown to heighten the proinflammatory response of myeloid cells *in vitro* and in animal models potentially through epigenetic mechanisms.⁸ This is also known as urate-induced training immunity. In our study, Mendelian randomization analysis did not identify significant causal effects of serum urate on DNA methylation, but it is conceivable that serum urate might act on other forms of epigenetic mechanisms, such as histone modification.⁹” (Discussion: page 21, paragraph 1)

Other comments.

1. In the context of the correlation with other metabolic traits it is important to point out that there is no robust evidence by MR for soluble urate being causal of these traits.

Response: We agree and added this interpretation in the Discussion section: “There has been no robust evidence supporting causal effects of serum urate on

cardiometabolic traits.^{10, 11} Instead, our observations are consistent with shared gene regulatory programs resulting in a common DNA methylation signature of serum urate and metabolic syndrome in whole blood” (Discussion: page 21, paragraph 2).

2. Line 253 'median of the mean age'.

Response: Each cohort reported the mean age within the cohort, which ranged from 39.7 years among African Americans in CARDIA to 75.4 years among European Americans in CHS (**Supplementary Table 1**). We selected the median of this mean age across cohorts for reporting. To clarify this, we have revised the text to read “the median of the average age within each cohort” (Results: page 8, paragraph 2).

3. In the mediation analysis there were only 3 independent variants reported at SLC2A9 in Tin et al (Table S5). Also, are these variants truly independent, as Tin et al 2019 used a genomic distance cut-off, there was no LD-clumping or conditional analysis. (And the SNPs in the current study were different than those reported in Tin et al 2019 - are they surrogates?)

Response: Tin *et al* 2019 reported 4 independent SNPs among individuals of European ancestry in the *SLC2A9* region (Nat Genet. vol 51, page 1462). Whereas the 183 loci based on the results from transethnic meta-analysis in Tin *et al.* were defined using a distance criterion, the independent SNPs for performing statistical fine-mapping were based on summary statistics from European ancestry individuals. The first step in statistical fine-mapping was the selection of independent SNPs (GCTA command: cojo-slct, collinearity <0.01) and resulted in the identification of the 4 independent SNPs (Tin *et al.* 2019, Supplementary Table 18, rs6825187, rs62286563, rs10017305, rs73224492). Thus, the 4 independent SNPs used in the current mediation analysis (see **Supplementary Table 12** of the current EWAS manuscript, containing these same SNPs) were selected based on LD rather than on genomic distance.

4. Something odd re references line 528.

Response: Thank you for the careful read. We have fixed the references.

5. Line 557, the phrase 'complexity of associations of SNPs in the SLC2A9 locus' did not make sense to me.

Response: We have revised the text to “Roles for SLC2A9 for urate uptake into cells in both secretory and reabsorption pathways may also help explain some of the heterogeneity of associations of SNPs at the *SLC2A9* locus.” (Discussion: page 23, paragraph 1).

6. Para beginning line 562. Of course the most associated genes are going to support the pathway analysis as they were used to generate the pathway analysis. This is a circular argument.

Response: We have revised the sentence to: “Terms or pathways enriched for urate-associated CpGs were mainly related to transmembrane transport of organic acids,” (page 23, paragraph 2).

7. Figures 2 and 3 - what is the LD measure used?

Response: Thank you for raising this point. The color in the bottom triangle in these figures represents the absolute values of the Pearson correlations between adjusted DNA methylation levels of the CpGs. The strength of the correlation can provide insight on the pattern of association between CpGs and urate. We used residuals of DNA methylation levels adjusting out blood cell type proportions and batch effects because these were the key technical covariates in the EWAS. We have added a color legend and related text in the legends of these figures (now **Figures 3 and 4**).

Reviewer #2 (Remarks to the Author):

In this interesting transethnic meta-analysis, Tin et al., have utilized 24 cohort studies to examine epigenome wide associations of serum urate. The authors have performed rigorous statistical analyses and used multiple layers of complementary evidence to support the validity of the findings. The authors have discussed on the complexity of these associations, biological plausibility of the results, and the associations of urate-associated methylation markers with other complex traits. The paper is well-written, clear description of methods and with noteworthy findings. I have a few minor comments.

Response: We thank the Reviewer for the positive feedback.

Line 254: It will be better to mention that the “Normative aging study (NAS)” only included men in the replication cohort. Other cohorts included both men and women.

Response: We have now noted that among the 8 replication cohorts, the Normative Aging Study (NAS) included only men (Results: page 8, paragraph 2. All page and paragraph numbers refer to the clean version of the manuscript).

Line 261: The authors have identified 140 significant CpGs (discovery dataset) associated with serum urate levels in the trans-ethnic meta-analysis including European, African American, South Asian and Sub-Saharan African Ancestries and 99 significant CpGs in the replication dataset. Did the authors evaluate whether the urate-associated methylation markers were significant across the ancestries included in the study, and whether there was any ancestry-specific DNA methylation marker? Additional information on the ancestry specific markers would be good to see in the supplementary information.

Response: Regarding heterogeneity across ancestries, we reported the I^2 heterogeneity measure from the meta-analysis that combined the results from the meta-analyses of individuals of European ancestry and of African Americans with results from individuals of South Asian ancestry (LOLIPOP only) and Sub-Saharan Africans (RODAM only) in **Supplementary Table 4**. Among the replicated CpGs, 32 had I^2 values $>50\%$. Of these, 90% (29 CpGs) had effect estimates in the same direction across all ancestries. To help visualize differences across ancestries, we are now providing forest plots of the ancestry-specific results at these 32 CpGs (**Supplementary Figures 2A to 2AF**). Regarding ancestry-specific CpGs, we now conducted additional epigenome-wide meta-analyses of all cohorts of European ancestry and of African-American ancestry. We added new supplementary tables to report the CpGs with p-value $<1.1E-7$ in each ancestry group when present, or with p-value $<1E-5$ otherwise (**Supplementary Tables 5, 6, 7 and 8**, respectively). The new supplementary tables are cited on in the Results section (page 9, paragraph 1).

Line 322: The causal effect estimates on serum urate (Table 1) results ranged from 0.16 to 0.30 mg/dl per standard deviation... instead of "0.15". Minor typo to correct.

Response: Thank you for pointing this out. We have adapted this sentence using the new results based on the suggestions from Reviewer 1 point 1. Among the promoter-associated CpGs, cg11266682 now remained significant with a causal estimate of 0.21 mg/dL per SD of DNA methylation beta value (Results: page 11, paragraph 2).

Line 324-325: Similar to the above sentence (Table 1), "effect sizes ranging from -0.38 to -0.51 mg/dl per SD.. instead of "-0.37". Minor typo to correct.

Response: To reflect the revised MR results, we updated the sentence to "ranging from -0.65 to -0.46 mg/dL per SD of DNA methylation beta value" (Results: page 11, paragraph 2).

Figure 2. Please include A and B in the main figures for PHGDH and SLC1A5. You have described separately as A and B in the legends but is missing in the figures.

Response: We have added the A and B labels in the main figure.

Figure 3, line 1026: Legend typo error for the “HM450K” annotation file.

Response: Thank you for the careful read. We have corrected the typo in the legend of this figure (now **Figure 4**).

Reviewer #3 (Remarks to the Author):

This is a remarkably dense paper, which I have had to read several times to understand what is going on. The authors are to be commended that all the analyses are stringently carried out and unbiased. However, I feel that more effort is necessary to communicate the biological messages from the results.

Methylation is a dynamic process that often reflects the regulatory state of particular loci. It is very rarely heritable, for example with imprinted loci. The authors are aware that apparent heritability may be attributed to underlying SNPS affecting transcription binding sites. Their investigation of SNP/DMR interactions is worthy but the train of the analysis is difficult to follow.

Response: Thank you for the positive feedback. We agree that heritability of DNA methylation at specific sites does not imply that DNA methylation is inherited across generations through the germline, but refers to the proportion of DNA methylation variance due to additive genetic effects. We have further clarified the concept of heritability in this context (Results: page 10, paragraph 2. All page and paragraph numbers refer to the clean version of the manuscript).

To help the reader to follow the train of analysis, we have now enhanced the workflow figure (former **Supplementary Figure 1**) and moved it to the main manuscript as **Figure 1** (cited in Results: page 8, paragraph 2).

The biological meat of the study is in table 4 of the supplementary information where CpG/DMRs and their associated genes are listed. There are some really interesting genes here that are unmentioned in the text. I would put an abbreviated form of this into the main paper and include columns with a brief description of putative gene function and an indicator of the presence of SNPs associated with the trait.

Response: In total, there were 81 unique nearest genes among the replicated CpGs. To prioritize the more interesting genes into a main table that fits on a journal page, we have now selected the 24 genes with any of the following 4 features: a) genes containing CpGs that are significantly associated with the expression of the corresponding gene in whole blood or monocytes, b) genes that map into known urate and/or gout loci from GWAS, c) genes with CpGs that were significant in MR analysis of urate or gout, and d) genes having CpGs associated with at least one of the metabolic syndrome traits reported in **Figure 8**. For these genes, we have now added a new **Table 1** that includes a brief description of gene function and indicators for the above 4 features. This table should also help readers with the navigation of the findings, in addition to the new workflow figure. **Table 1** is cited in Results, page 9, paragraph 1.

I would provide detailed maps, such as figure 2, of the most important of these loci, including recognised genomic features such as CpG islands and known regulatory elements, and the position of trait-associated SNPs.

Response: For plotting, we prioritized the genes with more than one replicated CpG that also showed significant associations with gene expression (*PHGDH*, *SLC1A5*, and *SLC2A9*). No other genes met these criteria (see new **Table 1**). To provide more information on regulatory genomic features, we have now added annotations for CpG islands and CpG position in the gene to **Supplementary Table 4**. Regarding urate-associated SNPs, three genes with replicated CpGs were annotated as known GWAS loci for serum urate (*SLC2A9*, *NBPF20/PDZK1* and *HRASLS2*, see new **Table 1**). We only found evidence for the relationship of genetic signals and DNA methylation at *SLC2A9*. To present the MR results, we plotted the meQTLs at the CpGs that were significant in the forward MR analysis (**Supplementary Figures 5A to 5D** for urate and **Supplementary Figure 6** for gout). For the other genes with replicated CpGs and mapping into known urate GWAS loci, there is no evidence that the genetic and DNA methylation signals are related.

I find the description of Mendelian Randomisation choked with jargon and really really difficult to understand. Some simple explanation would make the paper far more accesible to non-statistical readers.

Response: Thank you for this comment. We have added explanations on the basic assumptions of an MR study in the Methods section along with the ways that our selection criteria for genetic variants and analysis methods address the MR assumptions (page 30, paragraph 2).

A major problem is that the DNA for DMR identification came from peripheral blood, which is a mixture of different cell types with quite different functions and many cell-

specific DMRs. This means that DMRs associated with any of the metabolic syndrome traits may simply reflect changes in the elements of the white cell count. Some effort needs to be made to deconvolute this. At the simplest level this could be detecting correlations with WCC subsets (neutrophils, eosinophils etc. etc.), but the use of invariant CpG markers of cell lineages or WGCNA analyses are also possible

Response: We agree it is important to estimate associations between DNA methylation and a trait independent from blood cell type composition. A common and accepted practice is to adjust for blood cell type proportions that have been measured or imputed from markers of cell lineages.¹² Our study, as well as all EWAS of cardiometabolic traits used in our manuscript, have followed these established methods to minimize any effect of differential blood cell type proportions. **Supplementary Table 2** provides an overview of the study-specific procedures for measuring or estimating the proportions of white blood cell types. **Supplementary Table 19** reports the covariates, including cell type proportions, used in the EWAS of cardiometabolic traits.

As suggested by the Reviewer, we now performed additional checks by computing the Pearson correlations between blood cell type proportions and the residuals of the DNA methylation levels of the 17 urate-associated CpGs found to be associated with cardiometabolic traits using data from the participants of European ancestry in the ARIC study. The residuals were generated by adjusting out imputed blood cell type proportions and batch effects, technical covariates used in the EWAS analysis. As shown in **Reviewer Table 2**, the low correlations between cell type proportions and the residuals of DNA methylation support our approach that controlling for blood cell type proportions indeed allows for the detection of association between DNA methylation levels and traits independent of white blood cell proportions. We cannot exclude the possibility that inaccurate values of cell type proportions may generate biased estimates. However, the CpG associations used in this analysis of cardiometabolic traits were mostly generated from meta-analyses of multiple studies like ours, which serve to reduce biases from any one study. We have revised the Methods section to clarify the importance of controlling for cell type proportions in EWAS and pointed to **Supplementary Tables 2 and 19**, showing that this approach was chosen by all studies (Methods: page 27, paragraph 2; page 39, paragraph 1).

Reviewer Table 2. Pearson correlations between imputed blood cell type proportions and residuals of DNA methylation adjusting out batch effect and cell type proportions (technical covariates in the EWAS analysis) among participants of European ancestry (n=741) in the ARIC study. The blood cell type proportions were imputed using the Houseman algorithm.¹²

	B cell	CD4 T cell	CD8 T cell	Granulocyte	Monocyte	Natural Killer cell
cg03725309	0.004	-0.005	0.006	-0.003	-0.002	0.007
cg16246545	0.001	0.005	0.001	0.002	-0.005	-0.015

cg14476101	-0.002	0.004	0.007	-0.006	0.000	-0.005
cg19693031	0.003	0.008	-0.004	0.010	-0.013	-0.027
cg06690548	-0.001	-0.004	-0.017	0.010	-0.011	-0.011
cg18120259	0.016	0.003	0.008	-0.012	-0.008	0.019
cg21429551	-0.017	-0.010	0.000	0.001	0.011	0.006
cg22103219	-0.011	-0.015	-0.010	0.010	-0.001	-0.001
cg17061862	-0.012	-0.004	-0.005	0.005	0.004	-0.003
cg11376147	-0.003	-0.016	-0.007	0.006	0.020	-0.010
cg00574958	-0.002	0.004	-0.008	0.004	-0.007	-0.007
cg01243823	0.006	0.002	0.002	-0.010	0.024	0.006
cg26470501	-0.008	0.005	0.006	-0.013	0.013	0.009
cg22304262	0.006	0.012	0.001	-0.012	0.007	0.000
cg02711608	-0.005	0.000	-0.004	-0.002	0.021	-0.003
cg21766592	0.004	-0.002	-0.011	0.005	0.006	0.000
cg01881899	-0.002	0.006	0.018	-0.001	0.003	0.001

References

1. van Dongen J, *et al.* Genetic and environmental influences interact with age and sex in shaping the human methylome. *Nat Commun* **7**, 11115 (2016).
2. McRae AF, *et al.* Contribution of genetic variation to transgenerational inheritance of DNA methylation. *Genome Biology* **15**, R73 (2014).
3. Hannon E, *et al.* Characterizing genetic and environmental influences on variable DNA methylation using monozygotic and dizygotic twins. *PLoS genetics* **14**, e1007544-e1007544 (2018).
4. Min JL, *et al.* Genomic and phenotypic insights from an atlas of genetic effects on DNA methylation. *Nat Genet* **53**, 1311-1321 (2021).
5. Huan T, *et al.* Genome-wide identification of DNA methylation QTLs in whole blood highlights pathways for cardiovascular disease. *Nat Commun* **10**, 4267 (2019).
6. Burgess S, Butterworth A, Thompson SG. Mendelian randomization analysis with multiple genetic variants using summarized data. *Genet Epidemiol* **37**, 658-665 (2013).
7. Burgess S, *et al.* Guidelines for performing Mendelian randomization investigations [version 2; peer review: 2 approved]. *Wellcome Open Research* **4**, (2020).
8. Cabău G, Crișan TO, Klück V, Popp RA, Joosten LAB. Urate-induced immune programming: Consequences for gouty arthritis and hyperuricemia. *Immunol Rev* **294**, 92-105 (2020).

9. Crişan TO, *et al.* Soluble uric acid primes TLR-induced proinflammatory cytokine production by human primary cells via inhibition of IL-1Ra. *Ann Rheum Dis* **75**, 755-762 (2016).
10. Li X, *et al.* Serum uric acid levels and multiple health outcomes: umbrella review of evidence from observational studies, randomised controlled trials, and Mendelian randomisation studies. *BMJ* **357**, j2376 (2017).
11. Li X, *et al.* MR-PheWAS: exploring the causal effect of SUA level on multiple disease outcomes by using genetic instruments in UK Biobank. *Ann Rheum Dis*, (2018).
12. Houseman EA, *et al.* DNA methylation arrays as surrogate measures of cell mixture distribution. *BMC Bioinformatics* **13**, 86 (2012).

Reviewers' Comments:

Reviewer #1:

Remarks to the Author:

Thanks to the authors for addressing my points, all of which are addressed to my satisfaction. Some additional follow-up suggestions for improving the clarity:

1. Include the response regarding median number of meQTLs (para 2, page 2 of response letter) also in the paper, it helpful in putting the number of SLC2A9 meQTLs in context.
2. Please explain in the revised paper why ARIC was not included in the reverse MR.
3. Include the power calculation table in Supplemental material.

Reviewer #2:

Remarks to the Author:

Thank you for making the changes. No further comments.

Reviewer #3:

Remarks to the Author:

The authors have made a great effort to respond to the reviewers' suggestions, and the paper is much easier to read and to interpret. It is a formidable study that I am sure will become a landmark in the field. I am very happy with the detailed responses to the problems I identified.

NCOMMS-21-09981A; Response to Reviewers

Reviewer #1 (Remarks to the Author):

Thanks to the authors for addressing my points, all of which are addressed to my satisfaction. Some additional follow-up suggestions for improving the clarity:

1. Include the response regarding median number of meQTLs (para 2, page 2 of response letter) also in the paper, it helpful in putting the number of SLC2A9 meQTLs in context.

Response: Thank you for reviewing our revised manuscript. We agree with the Reviewer that this is helpful information, and have included it along with the section describing the selection of the meQTLs on page 29, paragraph 1.

2. Please explain in the revised paper why ARIC was not included in the reverse MR.

Response: We have included this explanation on page 31, paragraph 2.

3. Include the power calculation table in Supplemental material.

Response: We have included the power calculation table in the Supplementary Text on page 11, and refer readers to this calculation in the main manuscript on page 31, paragraph 1.

Reviewer #2 (Remarks to the Author):

Thank you for making the changes. No further comments.

Response: Thank you for reviewing our revised materials.

Reviewer #3 (Remarks to the Author):

The authors have made a great effort to respond to the reviewers' suggestions, and the paper is much easier to read and to interpret. It is a formidable study that I am sure will become a landmark in the field. I am very happy with the detailed responses to the problems I identified.

Response: Thank you for reviewing our revised materials, and for the positive assessment of our work.